# Amortizing intractable inference in large language models

**Edward J. Hu**[*], **Moksh Jain**[*], **Eric Elmoznino**
Mila – Quebec AI Institute, Université de Montréal
{edward.hu,moksh.jain,eric.elmoznino,...

**Younesse Kaddar**[∞]
University of Oxford
younesse.kaddar@chch.ox.ac.uk

**Guillaume Lajoie**[†], **Yoshua Bengio**[◇], **Nikolay Malkin**
Mila – Quebec AI Institute, Université de Montréal
...,guillaume.lajoie,yoshua.bengio,nikolay.malkin}@mila.quebec

## Abstract

Autoregressive large language models (LLMs) compress knowledge from their training data through next-token conditional distributions. This limits tractable querying of this knowledge to start-to-end autoregressive sampling. However, many tasks of interest—including sequence continuation, infilling, and other forms of constrained generation—involve sampling from intractable posterior distributions. We address this limitation by using amortized Bayesian inference to sample from these intractable posteriors. Such amortization is algorithmically achieved by fine-tuning LLMs via diversity-seeking reinforcement learning algorithms: generative flow networks (GFlowNets). We empirically demonstrate that this distribution-matching paradigm of LLM fine-tuning can serve as an effective alternative to maximum-likelihood training and reward-maximizing policy optimization. As an important application, we interpret chain-of-thought reasoning as a latent variable modeling problem and demonstrate that our approach enables data-efficient adaptation of LLMs to tasks that require multi-step rationalization and tool use.
Code: https://github.com/GFNOrg/gfn-lm-tuning.

## 1 Introduction

Autoregressive large language models (LLMs) trained on general-domain data are vast stores of world knowledge (Petroni et al., 2019). They are typically optimized by predicting a token given its preceding context; therefore, tractable inference over this knowledge is limited to sampling conditioned on a prefix. Many useful tasks, such as infilling (Zhu et al., 2019; Liu et al., 2019), generating text conditioned on length or lexical constraints (Hokamp & Liu, 2017; Hu et al., 2019), and finding the most likely sequence continuation, involve intractable inference in LLMs.

Such tasks are related to the problem of reasoning, which has been framed as one of probabilistic inference (Gershman & Goodman, 2014). Correspondingly, the linguistic expression of reasoning can be seen as inference over language. For example, we can interpret chain-of-thought reasoning (Wei et al., 2022; Kojima et al., 2022), a paradigm of reasoning in language models, as a problem of intractable posterior inference. Given a question-answer pair $(X, Y)$, we are interested in finding latent chains of thought – token sequences $Z$ that contribute the most to the conditional likelihood

$$p(Y \mid X) = \sum_Z p_{\text{LM}}(ZY \mid X) = \sum_Z p_{\text{LM}}(Y \mid XZ)p_{\text{LM}}(Z \mid X), \tag{1}$$

where $p_{\text{LM}}$ denotes the likelihood assigned to a sequence by a language model and apposition of variables (*e.g.*, $XZY$) denotes the concatenation of the token sequences.

While past work has relied on prompting and in-context learning to produce $Z$'s that lead to the correct $Y$, treating $Z$ as a hidden variable in a latent variable model (LVM) renders chain-of-thought reasoning a Bayesian inference problem (Fig. 1). For this LVM, the distribution we must sample from is the posterior $p_{\text{LM}}(Z \mid X, Y) = \frac{p_{\text{LM}}(XZY)}{\sum_{Z'} p_{\text{LM}}(XZ'Y)}$. Such sampling is intractable: while it is easy to evaluate $p_{\text{LM}}(XZY)$, the conditional distributions needed to sample $Z$ from $p_{\text{LM}}(Z \mid X, Y)$ one token at a time are not easy to compute.

---

[*]Equal contribution. [∞]Work done during internship at Mila. [†]CIFAR AI Chair. [◇]CIFAR Senior Fellow.

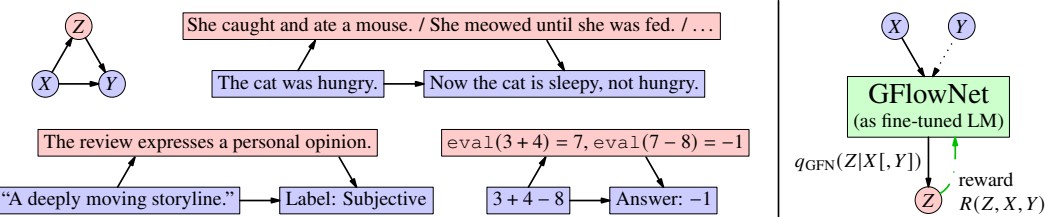

Figure 1: **Left:** Three problems of reasoning in language – sentence infilling, chain-of-thought reasoning, and problem-solving with external tool use – can all be seen as instances of the latent variable model at the top left, where an input ($X$) generates the output ($Y$) via a latent variable ($Z$). **Right:** We fine-tune an LLM to sample from the Bayesian posterior over $Z$, conditioning on $X$ and optionally on $Y$. If conditioned on $Y$, the trained policy can be used to sample diverse latent sequences (*e.g.*, for infilling, §4.2). If not conditioned on $Y$, the policy can sample $Z$, and thus predict $Y$, for inputs $X$ not seen during training (*e.g.*, for classification and multi-step reasoning, §4.3, 4.4). As shown in §4.4, modeling the full diversity of the posterior aids generalization.

A standard method to sample approximately from intractable posterior distributions is Markov chain Monte Carlo (MCMC), but it is difficult to craft good proposal distributions for multi-modal distributions over language data (Miao et al., 2019; Zhang et al., 2020a; Lew et al., 2023), and inference on a new input may be prohibitively slow. Alternatively, one can turn to reinforcement learning (RL) approaches such as proximal policy optimization (PPO; Schulman et al., 2017), where the language model is treated as a policy to be fine-tuned. However, these do not aim to model the full diversity of the distribution; instead, learned policies settle around a small number of modes. In both cases, issues with this mode collapse are exacerbated when the target distribution is misspecified, leading to the undesirable behavior of overoptimized samplers (Gao et al., 2023).

Amortized probabilistic inference – that is, training a model to approximate a distribution of interest – provides a principled, efficient, and potentially scalable way to draw samples from the distribution (Beal, 2003). One way to implement amortized inference for high-dimensional discrete data such as text is using generative flow networks (GFlowNets; Bengio et al., 2021), which are diversity-seeking reinforcement learning algorithms that train policies to sample objects (such as a token sequence $Z$) with probability proportional to a given reward function, such as the joint $p_{LM}(XZY)$.

In this work, we present a method that initializes the GFlowNet policy with a pretrained LLM and continues to train it with a reward objective that can be evaluated with the same LLM. The result is a different type of fine-tuning (FT) procedure for text generation that has a number of advantages, including improved sample diversity, data efficiency, and out-of-distribution generalization. GFlowNet fine-tuning makes the language model sample from the target distribution, enabling amortized inference in a number of applications (Fig. 1).

Leveraging this approach, we empirically demonstrate the possibilities and benefits of learning to sample from intractable distributions over text continuations, latent reasoning chains, and tool use sequences using GFlowNet fine-tuning. Notably, the diversity of samples from the models trained with GFlowNet fine-tuning is beneficial in Bayesian model averaging settings, such as when aggregating answers to questions obtained via multiple reasoning chains. For example, using a pretrained language model with 6B parameters, our method shows an absolute improvement of 10.9% over supervised fine-tuning on subjectivity classification with only 10 labeled examples (§4.3) and outperforms supervised fine-tuning and PPO by 63% on integer arithmetic with 50 demonstrations, with notable improvements in out-of-distribution generalization (§4.4). Moreover, the benefits of amortized inference allow us to efficiently sample from the fine-tuned model at scale. Our contributions include:

(1) A general algorithm for amortized sampling from intractable LLM posteriors.
(2) A probabilistic approach to fine-tuning LLMs to perform chain-of-thought reasoning.
(3) Empirical results on sequence continuation, natural language reasoning, integer arithmetic with tool use, and story infilling.

## 2 MOTIVATING EXAMPLE: GENERATING RANDOM NUMBERS WITH LLMS

We consider a simple task that highlights the limitations of reward-maximizing reinforcement learning (RL) methods in fine-tuning LLMs. For readers unfamiliar with RL, we refer to Sutton & Barto (2018) and include a glossary in §A.1 to define key terms used throughout this paper. The task involves generating random numbers from a given distribution when prompted *'The following is a random integer drawn uniformly between 0 and 100: '*. This task is a minimal instantiation of the problem we study in the paper: sample from a target distribution given an unnormalized density.

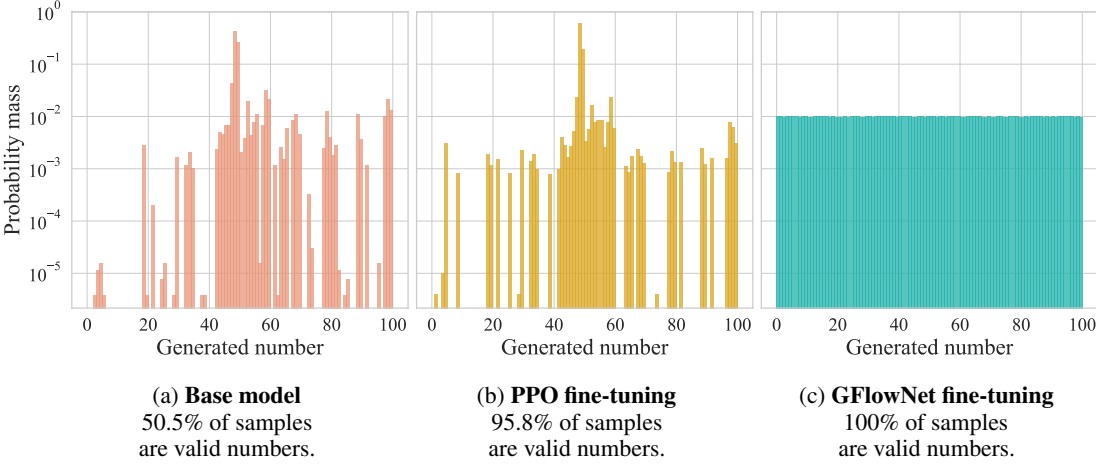

(a) **Base model**
50.5% of samples
are valid numbers.

(b) **PPO fine-tuning**
95.8% of samples
are valid numbers.

(c) **GFlowNet fine-tuning**
100% of samples
are valid numbers.

Figure 2: Empirical distributions of 512,000 integers from 1 to 100 generated by GPT-J fine-tuned with PPO (reward-maximizing; b) and GFlowNet fine-tuning (distribution-matching; c). Note the logarithmic *y*-scale.

Although the target distribution is tractable, making the task seemingly straightforward, it serves as a useful illustration of the behaviors of different fine-tuning methods.

Renda et al. (2023) found that pretrained LLMs perform quite poorly on this task: the distribution of numbers generated with the above prompt will be far from uniform (Fig. 2a shows an example using an instruction fine-tuned GPT-J 6B (Wang & Komatsuzaki, 2021)[1]). There may be many reasons for this, among them the effects of instruction fine-tuning and the LLM's possible bias towards numbers that are more frequent in the training data (*e.g.*, numbers starting with '1' are more frequent due to the properties of many natural data-generating processes (Benford, 1938)).

While reward-maximizing RL can teach the model to generate valid numbers (by penalizing outputs that are not numbers from 1 to 100), it would not resolve the distribution skew introduced during pretraining. Indeed, rewarding all valid integers equally leads to an expected gradient of zero for policy gradient methods. Fig. 2b shows that while most samples are valid numbers after PPO training, the distribution remains highly skewed.

Instead, we can take a principled approach by training the LLM to match the target distribution with a GFlowNet learning objective. Such an objective directly optimizes the likelihood of the model generating a number to be proportional to the reward for that number, which is the number's (potentially unnormalized) probability under the target distribution. When the policy is initialized as the pretrained LLM, the resulting distribution after GFlowNet fine-tuning is shown in Fig. 2c. Quantitatively, the KL divergence from the sampling distribution to the target (uniform) distribution decreases from 3.37 for the original LLM (on the support $[0, 100]$) to $9.75 \cdot 10^{-5}$ for the GFlowNet-fine-tuned model.

This example illustrates a general point: GFlowNet objectives provide a principled and flexible approach to fine-tuning LLMs to *match* a target distribution where reward-maximizing RL fails to. On this simple task, this distribution matching could also be achieved through supervised fine-tuning; however, this would require access to samples from the target distribution, which are unavailable in general (though not in this simple example). In the following sections, we further illustrate this point in non-trivial problems involving intractable inference, reasoning with latent variables, and tool use.

## 3 FINE-TUNING LLMS TO SAMPLE FROM INTRACTABLE DISTRIBUTIONS

We first describe how intractable inference emerges from interesting applications of LLMs, one of which is chain-of-thought reasoning seen through the lens of latent variable models, where the posterior distribution over the latent variable is intractable. We then discuss how GFlowNet objectives can be used to train amortized samplers to perform such intractable inference.

### 3.1 PROBLEM: INTRACTABLE INFERENCE IN LARGE LANGUAGE MODELS

Autoregressive language models decompose the distribution over sequences of tokens as a product of ordered conditionals: $p(w_{1:N}) = p(w_1)p(w_2 \mid w_1) \cdots p(w_N \mid w_{1:N-1})$. While this decomposition makes left-to-right sampling from the distribution tractable, sampling from other conditional distribu-

---

[1] We use the Instruct-GPT-J model available at 🤗 .co/nlpcloud/instruct-gpt-j-fp16.

Table 1: Objects in language posterior inference. Given a pretrained 'teacher' LM $p_{\text{LM}}$, we train a GFlowNet $q_{\text{GFN}}$ to sample the posterior $p(Z \mid X, Y)$. Amortization and generalization are achieved by making $X$, and optionally $Y$, an input to $q_{\text{GFN}}$.

| Object | Meaning | Example 1 (infilling) | Example 2 (subjectivity classification) |
|---|---|---|---|
| $X$ | cause / condition / question | *The cat was hungry.* | *A deeply moving storyline.* |
| $Z$ | mechanism / reasoning chain | *She ate a mouse.* | *This review expresses personal feelings.* |
| $Y$ | effect / answer | *Now the cat is sleepy, not hungry.* | *Answer: Subjective* |
| $p(Z \mid X)$ | conditional prior | | $p_{\text{LM}}(Z \mid X)$ |
| $p(Y \mid X, Z)$ | likelihood of effect given cause and mechanism | | $p_{\text{LM}}(Y \mid XZ)$ |
| $p(Z, Y \mid X)$ | conditional joint, reward for $Z$ | | $p_{\text{LM}}(ZY \mid X)$ |
| $p(Z \mid X, Y)$ | posterior (**intractable!**) | approximated and amortized by GFlowNet $q_{\text{GFN}}(Z \mid X[, Y])$ | |
| $q(Y \mid X)$ | posterior predictive / Bayesian model average | approximated as $\sum_Z q_{\text{GFN}}(Z \mid X) p_{\text{LM}}(Y \mid XZ)$, sampled as $Z \sim q_{\text{GFN}}(Z \mid X), Y \sim p_{\text{LM}}(Y \mid XZ)$ | |

tions is intractable. Various problems of language modeling can be viewed as sampling from such intractable conditionals in the distribution over sequences of an LLM; we give two such examples and related terminologies in Table 1. Some tasks we study in §4 are instances of these examples.

**Tempered and contrastive sampling.** In many applications (*e.g.*, translation, summarization, dialogue systems), one wishes to sample from a low-temperature distribution over sequences $Z$ conditioned on a prefix $X$, *i.e.*, $q(Z \mid X) \propto p_{\text{LM}}(XZ)^{1/T}$ for some temperature $T < 1$, as high-likelihood samples are more likely to be fluent or accurate continuations of $X$ (Tillmann & Ney, 2003). The limit of $T \to 0$ gives a distribution that is peaky on the most likely continuation. However, sampling from $q$, or finding its mode, is intractable, and it is common to resort to approximations, such as tempering the *tokenwise* conditional distributions or using beam search to search for a mode. A related problem is sampling a continuation with a correction for its unconditional likelihood, *e.g.*, $q(Z \mid X) \propto p_{\text{LM}}(XZ)^\alpha p_{\text{LM}}(Z)^\beta$ with $\beta < 0$ and $\alpha > 0$, where applications again resort to approximating the next-token conditionals of $q$ by tempering (Malkin et al., 2022b; Li et al., 2023).

**Infilling and reverse generation.** Infilling is the task of sampling a sequence of tokens conditioned on both its prior and subsequent context, which can be understood as sampling from the distribution $q(Z \mid X, Y) \propto p_{\text{LM}}(XZY)$, where $X$ and $Y$ are fixed. Reverse generation is a special case, where $X$ is an empty sequence. Besides being a meaningful task in its own right (Liu et al., 2019; Zhu et al., 2019; Donahue et al., 2020; Susanto et al., 2020; Lu et al., 2022a), infilling and reverse generation are key components of newly emerging methods of LLM prompting, such as when LLMs are tasked with optimizing their own instruction sequences or reasoning steps (Zhou et al., 2023; Sordoni et al., 2023; Xu et al., 2023). Current applications achieve this by resorting to hand-engineered instructions and inverted prompts.

**Constrained generation.** Sampling of text with constraints and penalties – for example, those on the presence or the absence of certain words or on the score of an auxiliary classifier evaluated on the text – can be understood as sampling from a distribution $q(Z) \propto p_{\text{LM}}(Z)c(Z)$, where $c$ is an externally specified constraint. Current approaches to the problem use tokenwise approximations (Liu et al., 2021), various problem-specific beam search and local search techniques (*e.g.*, Schmaltz et al., 2016; Hokamp & Liu, 2017; Hu et al., 2019; Sha, 2020; Lu et al., 2022b) or classifier-guided conditional generation approaches (*e.g.*, Yang & Klein, 2021; Meng et al., 2022).

## 3.2 REASONING THROUGH LATENT VARIABLES

Chain-of-thought reasoning (Wei et al., 2022; Kojima et al., 2022) helps LLMs solve complex problems by producing a reasoning chain before giving the final answer. LLMs pretrained on general domain data can learn to produce useful chains of thoughts given demonstrations, which are usually handcrafted or generated by prompting the LM. Interestingly, although the capacity for chain-of-thought reasoning only emerges in large language models, knowledge can also be extracted from smaller language models when they are carefully fine-tuned (Schick & Schütze, 2021).

Motivated by this, we connect chain-of-thought reasoning to the general problem of inference in latent variable models illustrated in Fig. 1. Here, reasoning can be seen as posterior inference: sampling from the posterior distribution over a string of tokens $Z$ conditioned on a prefix $X$ and a suffix $Y$, given an autoregressive language model $p_{\text{LM}}$. The posterior is defined as

$$p_{\text{LM}}(Z \mid X, Y) = \frac{p_{\text{LM}}(XZY)}{\sum_{Z'} p_{\text{LM}}(XZ'Y)} \propto p_{\text{LM}}(XZY). \tag{2}$$

Our goal is to train models to sample $Z$ from this posterior distribution. Intuitively, this allows us to sample likely reasoning chains that lead to the desired outcome $Y$. Although we take $Z$ to be a string

of tokens, the same formalism and the GFlowNet objectives apply to other structured latent objects, such as trees or sets of natural language statements, as long as one has access to a likelihood model $p(Y \mid XZ)$. While not investigated in this work, these generalizations could be important for formal reasoning and multi-step chains of inference. See, *e.g.*, Yao et al. (2023); Hao et al. (2023); Besta et al. (2024) for approaches to reasoning in language using tree- or list-structured state spaces.

A latent variable model of this form is useful when the marginal distribution $p_{\text{LM}}(Y \mid X)$ is harder to model than $p_{\text{LM}}(Z \mid X)$ and $p_{\text{LM}}(Y \mid XZ)$, *i.e.*, a difficult inference is broken down into a chain of easier ones. By training a model to match the Bayesian posterior $p_{\text{LM}}(Z \mid X, Y)$, we can learn to sample latent reasoning chains that increase the likelihood of producing $Y$ from $X$ via the sampled $Z$.

However, we can also fine-tune the language model $p_{\text{LM}}(Z \mid XY)$ itself to maximize the likelihood of data pairs $(X, Y)$ under the LVM. While it is generally intractable to directly maximize the data likelihood $p_{\text{LM}}(X, Y) = \sum_Z p_{\text{LM}}(XZY)$ because of the summation over $Z$, the (variational) expectation-maximization (EM) algorithm (Dempster et al., 1977; Beal, 2003; Koller & Friedman, 2009) can be used for this purpose. In the expectation step (E-step), we draw samples from the posterior over the latent variable $p_{\text{LM}}(Z \mid X, Y)$, which could come from an amortized sampler of $Z$. In the maximization step (M-step), we maximize the log-likelihood of the joint probability of the sampled latent variables $\mathbb{E}_{Z \sim p_{\text{LM}}(Z \mid X,Y)} \log p_{\text{LM}}(XZY)$ with respect to the parameters of the language model $p_{\text{LM}}$. This combination of amortized inference (learning to sample the chain of thought) and supervised fine-tuning (optimizing the language model with the 'supervision' involving $Z$ sampled from the amortized posterior) will be illustrated in one of our experiments (§4.3, Table 3).

### 3.3 Amortized inference with GFlowNet objectives

For inference in the latent variable model, we leverage the probabilistic framework of generative flow networks (GFlowNets; Bengio et al., 2021; 2023). Using notation from Malkin et al. (2022a), we briefly introduce relevant GFlowNet concepts pertaining to autoregressive sequence generation. Here, GFlowNets learn policies to sample sequences $Z = z_1 z_2 \ldots z_n \top \in \mathcal{Z}$ (where $\top$ denotes a stop symbol) from a distribution over the space of sequences $\mathcal{Z}$, given an unnormalized density (reward) $R : \mathcal{Z} \to \mathbb{R}_{>0}$. The generative process is the same as in autoregressive language models: generation begins with an empty string, and at the $i$-th step a token $z_i$ is sampled from a policy $q_{\text{GFN}}(z_i \mid z_{1:i-1})$, which is then appended to the sequence. This process continues until a stop symbol $\top$ is generated.

The marginal likelihood $q_{\text{GFN}}^\top(Z)$ of sampling a terminal state $Z = z_{1:n}\top$ is given by $\prod_{i=1}^{n} q_{\text{GFN}}(z_i \mid z_{1:i-1}) q_{\text{GFN}}(\top \mid z)$, where $z_{1:0}$ is understood to be the empty string. The goal of GFlowNet training is to fit a parametric policy $q_{\text{GFN}}(\cdot \mid \cdot; \theta)$ such that $q_{\text{GFN}}^\top(Z) \propto R(Z)$, *i.e.*, the likelihood of generating a complete sequence is proportional to its reward.

**Learning objective.** We use a modified version of the subtrajectory balance (SubTB; Madan et al., 2023) objective to account for trajectories being terminable at all states (Deleu et al., 2022). The objective for a sequence $Z = z_{1:n}\top$ is

$$\mathcal{L}(Z; \theta) = \sum_{0 \le i < j \le n} \left( \log \frac{R(z_{1:i}\top) \prod_{k=i+1}^{j} q_{\text{GFN}}(z_k \mid z_{1:k-1}) q_{\text{GFN}}(\top \mid z_{1:j})}{R(z_{1:j}\top) q_{\text{GFN}}(\top \mid z_{1:i})} \right)^2, \qquad (3)$$

For sequence generation tasks, the SubTB objective is equivalent to the path consistency objective (Nachum et al., 2017) in max-entropy RL (Haarnoja et al., 2017), which has been previously used in the context of text generation (Guo et al., 2021). See Appendix A.2 for further discussion.

**Training policy.** As the objective in Eq. 3 can be minimized to 0 for all trajectories $\tau$ simultaneously given enough model capacity, we can use trajectories sampled from *any* full-support distribution (training policy) to perform gradient descent on $\mathcal{L}(\tau; \theta)$ with respect to $\theta$. As the space we are sampling from is combinatorially large, it is important to have a training policy that can efficiently explore $\mathcal{Z}$. To this end, we compose the mini-batch during training using trajectories from three sources: (1) the policy $q_{\text{GFN}}$, (2) a tempered version of the current policy $q_{\text{GFN}}$ and (3) a replay buffer storing past trajectories. Replay buffers have been shown to be quite effective in improving GFlowNet training (Jain et al., 2022; Deleu et al., 2022; Shen et al., 2023).

**Parametrization, amortization, and generalization.** To sample the latent sequence $Z$ from the posterior defined in Eq. 2, we parametrize the GFlowNet policy as an autoregressive language model that samples the latent $Z$ one token at a time from left to right. By setting the reward $R(Z) = p_{\text{LM}}(XZY) \propto p_{\text{LM}}(Z \mid X, Y)$, we learn a sampler for the posterior at convergence.

As illustrated in Fig. 1, depending on the task, we can condition the GFlowNet policy on either $X$ or $X, Y$. In cases such as reasoning (§3.2), where there is only a single correct $Y$ for each $X$ and we interested in predicting $Y$ for unseen $X$ at test time, we can simply condition on $X$. In this case, the GFlowNet policy is simply a language model that generates $Z$ as a continuation of $X$. To be precise, we initialize $q_{\text{GFN}}$ as a copy of $p_{\text{LM}}$ that is conditioned on the prefix $X$, and then fine-tune[2] it with a GFlowNet objective. With this view, sampling $Z$ is an inverse problem: we need to infer $Z$ given a (conditional) prior $p_{\text{LM}}(Z \mid X)$ and an observation $Y$ under likelihood model $p_{\text{LM}}(Y \mid XZ)$.

Allowing the GFlowNet policy to explicitly take $X$ as input amortizes the sampling procedure and allows generalization to unseen $X$. In this sense, the GFlowNet is a Bayesian model (akin to a LM cascade (Dohan et al., 2022) or deep language network (Sordoni et al., 2023)), in which $Z$ are conditionally sampled 'parameters' that transform $X$ into $Y$. To predict the $Y$ for an unseen $X$, one performs Bayesian model averaging by drawing samples of $Z$ from $q_{\text{GFN}}(Z \mid X)$ followed by sampling from $p_{\text{LM}}(Y \mid XZ)$.

In tasks such as infilling (§4.2), however, the mapping from $X$ to $Y$ is one-to-many and $Y$ is available at test-time. Here, we are interested in $Z$ itself, rather than using it as an intermediate variable en route to generating $Y$. The GFlowNet policy thus has to be conditioned on both $X$ and $Y$. To achieve this, the policy is conditioned on a prompt that contains both $X$ and $Y$ (for example, see Appendix C).

## 4 EMPIRICAL RESULTS

We first validate GFlowNet fine-tuning on text generation, where we seek to find likely sentence continuation given a prompt (§4.1) or fill in a missing sentence in a story (§4.2). Then, we study reasoning tasks that benefit from chain-of-thought reasoning (§4.3) and external tool use (§4.4).

### 4.1 SENTENCE CONTINUATION

**Task description.** A natural application for autoregressive language models is that of sequence continuation: given a prompt, the model should generate a high-likelihood completion. In applications such as creative writing, we would like the continuations to be semantically diverse while still having a high likelihood under the language model. To demonstrate the benefits of GFlowNet fine-tuning, we consider the task of sampling the next sentence following a prompt.

Sampling autoregressively from the LM until a "." token is reached is unlikely to produce samples that have a high likelihood because the distribution over sentences has a fat tail. Existing approaches to generate sequence continuations include beam search and its variations (Vijayakumar et al., 2018; Shao et al., 2017), top-$k$ sampling (Fan et al., 2018), nucleus sampling

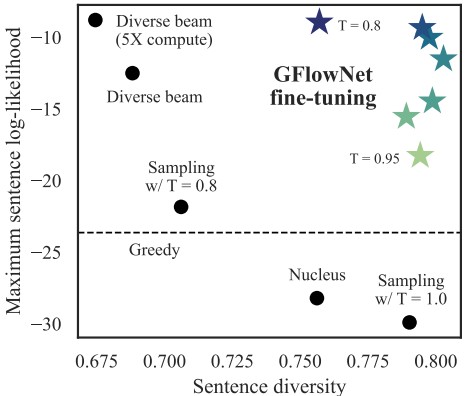

Figure 3: Maximum log-likelihood and diversity of continuations sampled for fixed prompts. GFlowNet fine-tuning (★) samples higher log-likelihood sentences while maintaining more sample diversity than the baselines (● and ---), even when they are given 5× the compute.

(Holtzman et al., 2019), tempered autoregressive sampling, and fine-tuning using importance sampling (Shih et al., 2023), among others. While useful, most of these methods are ultimately hand-crafted heuristics that leave room for improvement. Furthermore, some of these methods (*e.g.*, beam search) involve a computationally expensive search procedure, compared to a single pass of a learned inference model that amortizes over prompts. Our GFlowNet policy autoregressively samples the sequence until a period is sampled, indicating the end of the sentence. Given prompts $X$, the LM is fine-tuned to generate the continuations $Z$ from the tempered posterior by being trained with reward $R(Z) = p_{\text{LM}}(Z|X)^{\frac{1}{T}}$. When $T = 1$, the GFlowNet will trivially sample proportional to $p_{\text{LM}}$ without any fine-tuning, so we consider $0 < T < 1$ to focus on the likely continuations.

We consider a dataset of prompts from OpenWebText (Gokaslan et al., 2019) with a 1.5B param GPT-2 XL (Radford et al., 2019) as the base model. We draw 8 samples from the fine-tuned model conditioned on a fixed prompt, consider the maximum-likelihood sample under the LM, and report the average over the dataset of prompts. To measure the semantic diversity of the

---

[2]We use LoRA (Hu et al., 2022) instead of full fine-tuning for hardware efficiency in all experiments.

samples, we compute the mean pairwise cosine distance between the embeddings (from a pretrained encoder (Reimers & Gurevych, 2019)) of the generated samples and average it over the dataset. We compare to baselines that are commonly used for producing continuations from LMs at inference time (beam search, diverse beam search, nucleus sampling, autoregressive sampling, tempered autoregressive sampling, and greedy generation).

**Results.** Quantitative results are reported in Fig. 3 and empirical samples are shown in Appendix B. At lower temperatures, our method excels in generating high-likelihood sentences, outperforming the leading baseline, diverse beam search. If we increase the number of beams (and therefore compute) to 5× the number of samples produced by the GFlowNet, our performance remains comparable. Nevertheless, even in this scenario, the GFlowNet's generated samples exhibit notably higher diversity compared to diverse beam search and are on par with the best diversity-scoring benchmarks.

## 4.2 Infilling stories

**Task description.** Next, we consider the story infilling task, a special case of the general infilling problem (§3.1), where given the beginning $X$ and end $Y$ of a story, the goal is to generate the middle of the story $Z$ (Zhu et al., 2019). This is challenging for a language model sampled left to right since continuations $Z$ conditioned only on $X$ might be incompatible with the ending $Y$. We use the ROCStories corpus (Mostafazadeh et al., 2016), a dataset of short stories containing exactly 5 sentences each. Given the first 3 sentences and the last sentence, the goal is to generate the fourth sentence, which often involves a turning point in the story and is thus challenging to fill in.

As we expect the base model to contain the required knowledge, for this task we use a GPT-2 Large model (Radford et al., 2019) fine-tuned on the entire ROCStories training set as the base model. For evaluating the approach, we consider 900 samples from the dataset as training data to learn $q_{\text{GFN}}(Z|X,Y)$ and evaluate the similarity of the generated infills on a dataset of 100 unseen stories. Along with the GFlowNet-fine-tuned model, we also consider two baselines: prompting the model to infill the story and supervised fine-tuning on the same data. Further details are in Appendix C.

**Results.** To measure the similarity of the generated infills with the reference infills available in the dataset, we compute BERTScore (Zhang et al., 2020b), with DeBERTa (He et al., 2021) – which is correlated with human judgments –

Table 2: Evaluation of the generated infills.

| Method | BERTScore | BLEU-4 | GLEU-4 | GPT4Eval |
|---|---|---|---|---|
| Prompting | $0.081 \pm 0.009$ | $1.3 \pm 0.5$ | $3.2 \pm 0.1$ | 2.4 |
| Supervised fine-tuning | $0.094 \pm 0.007$ | $1.6 \pm 0.8$ | $3.7 \pm 0.4$ | 2.7 |
| GFlowNet fine-tuning | $\mathbf{0.184 \pm 0.004}$ | $\mathbf{2.1 \pm 0.2}$ | $\mathbf{4.2 \pm 0.7}$ | $\mathbf{3.4}$ |

along with BLEU-4 (Papineni et al., 2002) and GLEU-4 (better suited for sentences; Wu et al., 2016) metrics. Additionally, we also evaluate each method using GPT-4 as a judge. From our results summarized in Table 2, we observe that the infills generated by the model with GFlowNet fine-tuning are closer to the reference infills in the dataset than the baselines. By sampling from $p_{\text{LM}}(Z|X,Y)$, the GFlowNet is able to account for the ending while generating the infill, resulting in infills that link the beginning and the end of the story coherently. For further analysis and details see Appendix C.

## 4.3 Subjectivity classification

**Task description.** SUBJ (Pang & Lee, 2004) is a binary classification dataset for natural language understanding. It is a collection of movie reviews in which each review is labeled as *objective*, meaning that it references facts about the movie, or *subjective*, meaning that it expresses an opinion of the reviewer (see Table D.1 for examples). Given an unlabeled review, the model must predict whether it is objective or subjective. While supervised fine-tuning on the full dataset

Table 3: Test accuracy (%) on SUBJ using an instruct-fine-tuned GPT-J 6B.

| Method | Test accuracy (%) ↑ | | |
|---|---|---|---|
| Zero-shot prompting | 51.7 | | |
| | Training samples | | |
| | 10 | 20 | 50 |
| Few-shot prompting | $61.3 \pm 6.2$ | $61.8 \pm 5.4$ | $65.8 \pm 10.5$ |
| Supervised fine-tuning | $64.3 \pm 2.8$ | $69.1 \pm 0.8$ | $\mathbf{89.7 \pm 0.4}$ |
| GFlowNet fine-tuning | $71.4 \pm 1.8$ | $\mathbf{81.1 \pm 0.4}$ | $87.7 \pm 2.2$ |
| + Supervised fine-tuning | $\mathbf{75.2 \pm 1.8}$ | $78.7 \pm 1.6$ | $\mathbf{89.9 \pm 0.2}$ |

can achieve high test accuracy, we are interested in the low-data regime where we only have tens of labeled examples. We use the same instruction-tuned GPT-J 6B variant as in §2 for this experiment. Without any demonstrations, the model struggles with this task using the prompt in Table D.2 and achieves only 51.7% zero-shot accuracy.

This task is hard likely because it requires a latent reasoning step. A review could be considered objective because it analyzes the plot or facts about the movie, or it could be subjective because it expresses a personal opinion or makes a judgment. We denote the review $X$, the predicted subjectivity $Y$, and the latent reason $Z$. Then, we GFlowNet-fine-tune the LLM $q_{\text{GFN}}(Z \mid X)$, initialized with the

base model $p_{LM}$ to match the Bayesian posterior over rationales in Eq. 2. At test time, $q_{GFN}(Z \mid X)$ generates 10 latent rationales ($Z$'s) for an unseen $X$. The LLM $p_{LM}$ then autoregressively samples from $p_{LM}(Y \mid XZ)$ to produce 10 answers, the majority vote of which becomes the final prediction.

This posterior inference corresponds to the E-step in the EM algorithm, where the posterior $p_{LM}(Z \mid X, Y)$ is defined in Eq. 2. Further, as described in §3.2, we can take an M-step by updating $p_{LM}$ to maximize $\log p_{LM}(XZY)$ over a collection of $Z$'s sampled from the amortized posterior $q_{GFN}$. This is equivalent to applying supervised fine-tuning after GFlowNet fine-tuning.

**Results.** We present few-shot prompting and supervised fine-tuning with LoRA as baselines. In few-shot prompting, we prepend 0, 10, 20, or 50 training examples to each test example using the prompt shown in Table D.2. We randomly shuffle few-shot demonstrations and report the mean and variance in Table 3. In supervised fine-tuning, we directly maximize $\log p_{LM}(Y \mid X)$ over the same 10, 20, or 50 $(X, Y)$ pairs. The variance is over model initialization and batch order. All entries except zero-shot prompting are aggregated over 3 random seeds. See Appendix D for experiment details. GFlowNet fine-tuning consistently outperforms supervised fine-tuning in the low-data regime, as shown in Table 3. In some cases, performing supervised fine-tuning on top, which corresponds to running one step of the EM algorithm, further improves the performance.

## 4.4 SOLVING ARITHMETIC PROBLEMS STEP BY STEP

**Task description.** Arithmetic reasoning is a fitting benchmark to evaluate reasoning abilities of large language models as it requires multistep reasoning and correctness is easy to evaluate (Cobbe et al., 2021). While the distribution of pretraining and fine-tuning data (Magister et al., 2023; Lee et al., 2023; Luo et al., 2023) and prompting choices (Imani et al., 2023) play a critical role in their arithmetic abilities, LLMs are susceptible to poor generalization by learning 'shortcuts' to reasoning (Dziri et al., 2023). We consider a simple integer arithmetic task (Fig. 1), with a general pretrained base model, rather than a one pretrained on mathematical

Table 4: Test accuracy (%) on an integer arithmetic task with addition and subtraction using a GPT-J 6B model. Training data only include samples with 3 or 4 operands.

| | | Number of Operands | | |
| --- | --- | --- | --- | --- |
| | | In-distribution | | OOD |
| Method | | 3 | 4 | 5 |
| $k$-shot CoT | $k = 0$ | 10.2 | 6.4 | 3.2 |
| | $k = 3$ | 15.8 ± 3.1 | 11 ± 1.7 | 5.4 ± 0.2 |
| | $k = 5$ | 20.4 ± 10.4 | 17.6 ± 0.6 | 6.6 ± 1.1 |
| | $k = 10$ | 26.5 ± 1.4 | 15.2 ± 1.7 | 8.9 ± 1.9 |
| | $k = 20$ | 35.5 ± 1.9 | 21 ± 1.4 | 10.5 ± 0.9 |
| Supervised fine-tuning | | 72.1 ± 1.3 | 19.6 ± 2.2 | 12.8 ± 5.7 |
| PPO | | 30.6 ± 4.1 | 13.7 ± 4.1 | 5.6 ± 3.1 |
| GFlowNet fine-tuning | | **95.2 ± 1.3** | **75.4 ± 2.9** | **40.7 ± 9.1** |

tasks (Jelassi et al., 2023). To avoid the pitfalls of symbolic calculations with language models, we adopt the tool use setting (Schick et al., 2023), where the model is equipped with a calculator that can perform parts of the computation, implemented as in Cobbe et al. (2021): when the model outputs '=' the expression preceding it is evaluated and appended to the sequence. To prevent the model from "cheating", we limit the calculator to evaluate only two-term expressions. Consequently, reasoning here involves learning to plan using a tool with limited capabilities (Hao et al., 2023).

For training, we use a synthetic dataset of arithmetic expressions, limited to addition and subtraction. Following Zelikman et al. (2022), we use a small set of 50 demonstrations $(X, Z, Y)$ to seed the replay buffer in addition to 1000 examples $(X, Y)$. We use the same instruction-tuned GPT-J as in §4.3 as the base model. Further details are in Appendix E. We report the accuracy on two types of examples: (1) unseen in-distribution expressions (3 or 4 operands) and (2) longer out-of-distribution expressions (5 operands). As baselines, we consider zero-shot chain-of-thought prompting, $k$-shot prompting, supervised fine-tuning on the tool use sequences, and fine-tuning with PPO (Schulman et al., 2017). For all methods, we enable tool use and limit the model to generate only numbers and operators.

**Results.** From the results summarized in Table 4, the base model performs poorly even with chain-of-thought prompts. Including examples in context improves the performance considerably, with monotonic improvements as the number of examples increases. Supervised fine-tuning improves the performance significantly on the in-distribution examples, but the model still struggles to generalize on the out-of-distribution examples. Fine-tuning with PPO results also yields poor performance, caused in part by the poor calibration of the base reward model, *i.e.* it cannot distinguish good rationales from bad ones. Even though the sequences generated with PPO (illustrated in Appendix E) have high rewards, they are spurious and do not even define valid calls to the tool.

Such overoptimization to a misspecified reward is a widely noted issue in LLMs trained with RL (Gao et al., 2023). On the other hand, by matching the entire distribution, GFlowNet fine-tuning avoids collapsing to a single mode of the reward, thereby being robust to the misspecification of

the reward (Eysenbach & Levine, 2022) and achieving significantly better performance on in and out-of-distribution examples. See Appendix E for additional results and analysis.

## 5    FURTHER RELATED WORK

**Sampling from intractable marginals.**    Beyond the approximations mentioned in §3.1, sampling from intractable posterior distributions given by pretrained models for tasks such as infilling and constrained generation has been an object of study. Miao et al. (2019); Zhang et al. (2020a) use MCMC for these problems, Malkin et al. (2021) used a variable-neighborhood ascent for finding modes, and a sequential Monte Carlo approach was recently proposed by Lew et al. (2023). Others have studied the problem with *masked* language models, using them to perform variants of Gibbs sampling (Wang & Cho, 2019; Goyal et al., 2022; Yamakoshi et al., 2022) and recovering marginal distributions over small sets of tokens (Torroba Hennigen & Kim, 2023).

**GFlowNets.**    GFlowNets (Bengio et al., 2021) were originally proposed to learn policies for sampling discrete compositional objects from an unnormalized reward distribution, motivated by the need to sample diverse high-reward objects in scientific discovery (Jain et al., 2023), in particular, for biological sequence generation (Jain et al., 2022). The interpretation of GFlowNets as variational inference algorithms (Malkin et al., 2023; Zimmermann et al., 2023) makes them appropriate for sampling Bayesian posterior distributions over structured objects (*e.g.*, Deleu et al., 2022; 2023; van Krieken et al., 2023; Hu et al., 2023).

**Chain-of-thought reasoning in LLMs.**    In recent work on classification and completion with language models, the latent reasoning chain $Z$, in the notation of §3.1, is called a 'chain of thought' (Wei et al., 2022). The chain of thought is typically generated by conditioning the language model on $X$ with the use of specialized demonstrations or prompts (Kojima et al., 2022), with no guarantee of sampling the posterior accurately. Related to our Bayesian formulation, Wang et al. (2023b) noted that appropriately aggregating the conclusions $Y$ from several latent chains $Z$ improves predictive performance. In Xu et al. (2023); Zhou et al. (2022), a posterior over latent token sequences is sampled using MCMC, while Zelikman et al. (2022) propose *fine-tuning* on successful (high-reward, in our language) chains of thought, which achieves reward maximization but gives no guarantee of diversity. In concurrent work, Phan et al. (2023) use MCMC to sample chains-of-thought in problems with binary feedback. We expect these methods to generalize poorly to difficult exploration problems, while GFlowNet fine-tuning takes advantage of generalizable structure in the posterior and has a goal of sampling the full posterior over latent reasoning chains.

## 6    CONCLUSION

The knowledge compressed in LLMs is crucial for tasks such as infilling and constrained generation, but querying this knowledge involves sampling from intractable posterior distributions. We propose to use GFlowNet objectives to train LLMs to sample from such posterior distributions. Empirical results show that GFlowNet fine-tuning finds a better fidelity-diversity trade-off for text generation and also improves sample efficiency and generalization on downstream tasks compared to maximum-likelihood training or reward-maximizing policy optimization. As an amortized inference algorithm, our method converts computation into better test-time performance without additional data.

Future work should investigate transfer and generalization across tasks, in particular, building a 'universal reasoner' as a model $q(Z \mid X)$ shared between $X$ from different tasks, as was recently considered by Wang et al. (2023a). One should investigate the benefit of using a better knowledge model, *e.g.*, a more capable base LLM, as a starting point for GFlowNet fine-tuning. The ability to draw multiple samples from a GFlowNet can also be used to quantify epistemic uncertainty. Finally, we adopt the GFlowNet formalisms with the perspective of generalizing to latent variables $Z$ with a richer generative process than left-to-right sampling. We hope that the GFlowNet paradigm will enable more flexible reasoning with LLMs in the future: extending probabilistic programming with language variables (Beurer-Kellner et al., 2023), using structured chains of thought (Yao et al., 2023; Besta et al., 2024), and extending to program synthesis and planning with world models.

**Limitations.**    Due to resource constraints, our experiments use models up to 6B parameters, but we expect the conclusions to hold for larger models. In fact, our method can potentially benefit larger models more: it is harder to optimize a larger model with maximizing objectives on a small amount of data. As with any on-policy method, exploration, especially in problems with more complex latents, remains an open problem. Furthermore, our method improves inference but not the knowledge in the LM. Issues such as hallucination or miscalibration, which are closely related to the knowledge representation, are thus not addressed.

## ETHICS STATEMENT

While we foresee no immediate negative societal consequences of our work, we hope that future researchers who build upon it will, as we have, bear in mind the potential of LLMs – and in particular of human-like reasoning in LLMs – to be used both for good and for harm.

Research areas in safe and explainable AI that can benefit from GFlowNet fine-tuning include (1) interpretability of LLMs' reasoning processes and (2) fine-tuning with human feedback or an external reward, where diverse sampling can help prevent 'reward hacking' and overfitting to a misspecified target.

## REPRODUCIBILITY

We discuss the details of the proposed algorithms in §3.3 and provide all the implementation details and hyperparameters for the experiments in the main paper and appendix. Code for our experiments is available at https://github.com/GFNOrg/gfn-lm-tuning.

## ACKNOWLEDGMENTS

The authors are grateful to Bonaventure Dossou and Salem Lahlou for their help in the early stages of this project. We also thank Robert Hawkins, Arian Hosseini, Zhen Wang, and Anirudh Goyal for valuable discussions and suggestions of related work.

GL acknowledges funding from CIFAR, Samsung, and a Canada Research Chair in Neural Computation and Interfacing.

YB acknowledges funding from CIFAR, NSERC, IBM, Intel, Genentech, and Samsung.

The research was enabled in part by computational resources provided by the Digital Research Alliance of Canada (https://alliancecan.ca), Mila (https://mila.quebec), and NVIDIA.

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

# A    ADDITIONAL BACKGROUND

## A.1    GLOSSARY OF RL FOR LLMS

We provide definitions for key terms used throughout the paper, with a focus on their relevance to our setting of fine-tuning large language models (LLMs).

**Reinforcement learning.**    Reinforcement learning (RL) is a branch of machine learning concerned with how agents should take actions in an environment to maximize cumulative rewards. In our context, RL is used to fine-tune the decision-making process of LLMs to improve their performance on specific tasks. For a more comprehensive overview, we refer readers to Sutton & Barto (2018).

**Policy.**    In RL, a policy is a strategy that defines the behavior of an agent by mapping states of the environment to actions. In our setting, a policy dictates how the language model generates text sequences based on the current context and learned parameters.

**Reward.**    A reward is a signal that evaluates the quality of an action taken by an agent in a particular state. In the fine-tuning of LLMs, rewards are used to guide the model towards generating more desirable text, such as more accurate predictions or more coherent continuations. More specifically, in the context of GFlowNets, the reward corresponds to the unnormalized posterior probability, and the GFlowNet aims to match it by learning a policy that generates samples proportional to their reward.

**Matching a distribution.**    Matching a distribution involves training a model to approximate a target probability distribution. In our work, this concept is applied to fine-tune LLMs so that their generated text matches the desired characteristics, such as adhering to a particular style or content constraint.

**Policy gradient methods.**    Policy gradient methods (such as PPO) are a subset of RL algorithms that optimize a policy by computing gradients of the expected reward with respect to the policy parameters. In the context of LLMs, these methods are used to fine-tune the model's parameters to increase the likelihood of generating high-reward text sequences.

## A.2    LEARNING OBJECTIVE

We use the subtrajectory balance learning objective for training GFlowNets (Madan et al., 2023). In the notation of that paper, with a forward policy $P_F$, backward policy $P_B$ and state flow function $F$, the objective over a partial trajectory $\tau = s_m \rightarrow \cdots \rightarrow s_n$ is defined as follows

$$\mathcal{L}_{SubTB}(Z; \theta) = \left( \log \frac{F(s_m) \prod_{i=m}^{n-1} P_F(s_{i+1}|s_i)}{F(s_n) \prod_{i=m}^{n-1} P_B(s_i|s_{i+1})} \right)^2 \tag{4}$$

In the case of autoregressive generation of a sequence of tokens in a fixed order (left-right), the generative process is a tree, so there is only a single path to each state, and each state has a single parent. Thus, $P_B(s|s') = 1$ trivially. Additionally, since each state is a valid terminable state, we can incorporate the modification to account for this from Deleu et al. (2022). Specifically, note that at convergence we have $R(s_n^\top) = F(s_n)P_F(\top \mid s_n)$. Using this, we can simply substitute $F(s_n) = R(s_n^\top)/P_F(\top \mid s_n)$ in Eq. (4). This allows us to avoid parameterizing a flow function separately, reducing additional complexity. The only learned object is now the forward sampling policy $P_F$, which we refer to as $q_{GFN}$. Summing this over all partial trajectories in a trajectory with equal weight ($\lambda = 1$), we get the final learning objective in Eq. (3).

# B    SENTENCE CONTINUATION

**Additional details.**    We choose sentences as the level of granularity for our sequence continuation task because they are a natural unit of generation in language. Moreso than individual words, sentences are analogous to whole thoughts, reasoning steps, etc. because the compositional rules of syntax operate at the level of sentences. Whereas the meaning of a word is often ambiguous and context-dependent, the meaning of a sentence tends to be more self-contained.

A naive solution to the task is to simply sample autoregressively from the LM until a "." token is reached, marking the end of the sentence. In practice, however, this is unlikely to produce samples that have high likelihood because the distribution over sentences has a long tail: a vast number of sentences have small but non-zero probability, and collectively account for a substantial portion of the total probability mass. Instead, it would be desirable to sample from a low-temperature distribution over sentences, so that the distribution becomes more sparse and highly probable sentences are

sampled more often. However, this in itself is a difficult distribution to sample from, and it is different from simply sampling autoregressively from the LM with a lower temperature applied to the distributions over the next words. For instance, as the temperature approaches 0, the desired distribution over sentences should converge to an argmax and produce the single most likely next sentence. However, if we try to apply the temperature to the distribution over the next words and sample autoregressively, then as the temperature approaches 0, the resulting policy will greedily construct a sentence by sequentially picking the most likely next word, which is unlikely to produce the highest probability sentence.

Our GFlowNet sampling policy parametrizes a distribution $p_{\text{GFN}}(w_{i+1}|w_{1:i})$ using the same architecture as the LM $p_{\text{LM}}(w_{i+1}|w_{1:i})$, and at the start of training, it is initialized with the same weights. The initial state of the GFlowNet consists of a prompt $w_{1:k}$ that conditions generation (*i.e.*, the text whose next sentence we are trying to sample). Each subsequent state $w_{1:k+i}$ is an extension of this text and is obtained by sampling autoregressively from $p_{\text{GFN}}$. This process terminates when a period "." is sampled, indicating the end of the next sentence. If a predefined maximum sentence length is reached, we force a "." action to manually terminate the generation.

The GFlowNet policy is trained to sample sentences in proportion to their tempered probability under the original LM given the prompt. Denote the length of the prompt by $k$, the length of the sampled sentence by $m$, and the temperature by $T$. Then, the reward is:

$$R(w_{1:k+m}) \stackrel{\text{def}}{=} p_{\text{LM}}(w_{k+1:k+m}|w_{1:k})^{\frac{1}{T}} = \Big( \prod_{i=1}^{m} p_{\text{LM}}(w_{k+i}|w_{1:k+i-1}) \Big)^{\frac{1}{T}} \tag{5}$$

The intuition behind this reward is that we are assuming that the LM can reliably assign a high likelihood to high-probability sentences, that we wish to preferentially sample over low-probability ones. If we were to set $T = 1$, then the solution for the GFlowNet would be to sample proportionally to $p_{\text{LM}}$ (which it is initialized to). The preferential sampling of high-probability sentences is therefore obtained by setting $0 < T < 1$.

To run our experiment, we obtained a dataset of 1000 prompts from OpenWebText (Gokaslan et al., 2019) that were each 1-3 sentences long, 50 of which were used for validation. Our LM consisted of a pretrained 1.5B parameter GPT2-XL (Radford et al., 2019), and our GFlowNet was a copy of this model that was fine-tuned in a lower-dimensional space of 80M parameters using LoRA (Hu et al., 2022).

**Additional results.**   We show examples of empirical next-sentence samples generated by our model in Table B.1 compared to baselines. The model was trained using a reward temperature of 0.875, which achieves a good balance between log-likelihood and diversity.

## C   INFILLING STORIES

**Additional details.**   See Table C.1 for training examples from the subset of the ROC Stories dataset used for the task. To condition the model on $X$ and $Y$, as well as for the prompting baseline, we use the following prompt:

```
"Beginning: {X}\n End: {Y}\n Middle: "
```

An assumption we make in this paper is that the base language model already contains the knowledge required for the task, and the goal is to perform inference over this knowledge. However, for this task, none of the pretrained base models were good at assigning high likelihoods to plausible stories. Thus, we fine-tuned GPT-2 Large model (Radford et al., 2019) on the entirety of the stories dataset and used this fine-tuned model as the reward model. This was done with full fine-tuning using the `trl` library (von Werra et al., 2020). We trained for 20 epochs with a batch size of 64 and 32 gradient accumulation steps and a learning rate of 0.0005.

We detail the hyperparameters used for training GFlowNets in our experiments in Table C.2. During training, we sample $(X, Y)$ from the dataset and then sample (batch size) $Z$s for every $(X, Y)$, before using $p_{\text{LM}}(XZY)$ as the reward. We use the replay buffer described in §3.3 and seed it with rationales from the dataset. We linearly anneal the reward temperature, the temperature of the behavior policy during training, and the learning rate during warmup. For supervised fine-tuning, we use a batch size of 256 and train for 10 epochs with a learning rate of 0.0001 with `trl` and LoRA. At test time, we sample 1024 infills for each example in the test set from all the models at temperature 0.9, and average over 10 such draws.

Table B.1: Empirical examples for sequence continuation. Sentences generated from GFlowNet fine-tuning tend to be more reasonable than autoregressively generated samples from the LM with temperature 1.0 and tend to be more diverse than samples generated from diverse beam search.

| | |
|---|---|
| Input Prompt: | The matching campaign starts on Friday and will continue through midnight on Monday. |
| Sampling w/ $T = 1.0$: | (1) Participate with a fairly low \$1 donation so we can motivate more volunteers on Sunday afternoon. (2) However, the information regarding Cutler's suspicious death may not become widely known until early in the week. |
| Diverse beam search: | (1) If you are interested in participating in the matching campaign, you can do so by clicking here. (2) There is no limit to the number of times you can enter. |
| **GFlowNet fine-tuning**: | (1) If you are interested in signing up you can do so here. (2) Please share. |
| Input Prompt: | I want hockey to become a huge thing in Houston. |
| Sampling w/ $T = 1.0$: | (1) We'll be here in Texas from late November till end March. (2) To that end, I've been following the Houston Aeros (Houston Dynamo's minor-league affiliate) and their AHL affiliate (the Texas Stars). |
| Diverse beam search: | (1) That's why I'm here. (2) That's why I'm doing this. |
| **GFlowNet fine-tuning**: | (1) This is something I've always wanted. (2) When I was a teenager in middle school, I went to the Ice Arena in University of Houston and loved it. |
| Input Prompt: | That incident got a lot of attention in part because it was captured on video. Israel said he recorded what happened at the synagogue, and made it public, to document it and leave no doubt about what transpired. |
| Sampling w/ $T = 1.0$: | (1) He blogged about it here as well. (2) Israeli TV stations broadcast the video before it aired in Israel, as the country's rule requires. |
| Diverse beam search: | (1) However, there is no evidence that he did so. (2) However, there is no video of what happened inside the synagogue. |
| **GFlowNet fine-tuning**: | (1) He is on tape doing all of it. (2) It's a message countries can use to deter future attacks. |
| Input Prompt: | The Rolling Stones in Concert has since been released solely by ABKCO Records. The band would remain incensed with Klein for decades for that act. Klein died in 2009. |
| Sampling w/ $T = 1.0$: | (1) Actually art has become as obscene as investment banking. (2) Some believe he shot himself in the chest. |
| Diverse beam search: | (1) The Rolling Stones, however, would not be deterred. (2) The Rolling Stones would go on to release their own version of The Rolling Stones in Concert. |
| **GFlowNet fine-tuning**: | (1) He received a lifetime achievement award from the Jazz Times. (2) Sometimes it seems like we are destined to repeat our own mistakes. |

Table C.1: Examples of training samples for the stories infilling task.

| Beginning ($X$) | Middle ($Z$) | End ($Y$) |
|---|---|---|
| I was going to a Halloween party. I looked through my clothes but could not find a costume. I cut up my old clothes and constructed a costume. | I put my costume on and went to the party. | My friends loved my costume. |
| Allen thought he was a very talented poet. He attended college to study creative writing. In college, he met a boy named Carl. | Carl told him that he wasn't very good. | Because of this, Allen swore off poetry forever. |

Table C.2: Hyperparameters for the story infilling task.

| | |
|---|---|
| LoRA rank | 64 |
| LoRA scaling factor | 16 |
| LoRA dropout | 0.1 |
| Batch size | 64 |
| Gradient accumulation steps | 16 |
| Learning rate | 0.0001 |
| Optimizer | AdamW |
| $P_F$ temperature max | 2 |
| $P_F$ temperature min | 0.5 |
| Reward temperature start | 1.1 |
| Reward temperature end | 0.85 |
| Reward temperature horizon | 100 |
| Buffer capacity | 25 |
| Number of training steps | 1000 |
| Evaluation temperature | 0.8 |
| Maximum length | 25 |
| Minimum length | 5 |

**Analysis.** Table C.3, C.4, C.5, C.6, C.7, C.8 illustrate some examples of the infills generated with the prompting baseline, supervised fine-tuned and GFlowNet fine-tuned models on the test examples.

For evaluating the infills we generate a rating for the stories with infills from each of the methods based on the coherence of the story. We average the rating over 10 infills sampled from each method and average it over all the 100 test examples. The prompt used for evaluation is the following, adapted from Liu et al. (2023). Stories with infills generated by the GFlowNet fine-tuned model on average receive a much higher rating than the baselines. The average rating for the reference infills is **4.3** which should be viewed as an upper bound, as the stories may potentially be present in the GPT-4 training data.

```
You will be given a short story.

Your task is to rate the story on one metric.

Please make sure you read and understand these instructions
carefully. Please keep this document open while reviewing,
and refer to it as needed.

Evaluation Criteria:

Coherence (1-5) - the collective quality of all sentences.
The story should be well-structured and well-organized.
The story should not just be a heap of related information,
but should build from sentence to a coherent narrative.
The story should also be realistic and all the sentences
when put together should make sense.

Evaluation Steps:

1. Read the story carefully. Check if the story is coherent
and all the sentences follow a logical order.
2. Assign a score for coherence on a scale of 1 to 5,
where 1 is the lowest and 5 is the highest based on the
Evaluation Criteria. If there are grammatical errors
or logical inconsistencies,
simply assign a lower score and do not elaborate on the reasoning.

Example:

Randy had recently separated from his wife.
He felt very lonely and sad all the time.
He considered trying to patch things up with Vera, his ex-wife.
He decided to get a puppy instead.
His decision made him happy and he no longer felt sad.

- Coherence: 5

Randy had recently separated from his wife.
He felt very lonely and sad all the time.
He considered trying to patch things up with Vera, his ex-wife.
Eventually Randy missed Vera very much and developedWhile making
memories with
His decision made him happy and he no longer felt sad.

- Coherence 1
```

```
Story:

{{Story}}

Evaluation Form (scores ONLY):

- Coherence:
```

## D  SUBJECTIVITY CLASSIFICATION

**Additional details.**  See Table D.1 for some training examples in the SUBJ dataset. We use the prompts in Table D.2 for all baselines.

We run GFlowNet fine-tuning for 1000 steps with a linear warmup over 200 steps, a fixed learning rate of 0.0005, and a batch size of 512 samples; see Table D.3 for all the hyperparameters used. Each batch consists of 8 queries ($X$'s), randomly drawn with replacement. We then sample 64 rationales ($Z$'s) for every $X$, before using $p_{\text{LM}}(ZY \mid X)$ as the reward. We also use the replay buffer described in §3.3 and seed it with potential rationales generated from $p_{\text{LM}}$. For supervised fine-tuning, both on its own and on top of GFlowNet fine-tuning, we use a batch size of 256 with 8 queries, randomly drawn with replacement. We train for 100 steps with a linear warm-up over 20 steps and a constant learning rate. We sweep the learning rate in [0.001, 0.0001] and report the best performance. The reward temperature is annealed from 1.2 down to 1 over the first 150 steps of training.

**Additional results.**  To encourage exploration in the space of $Z$, we inverse-prompt the model to generate potential chains of thoughts and store them in a replay buffer at the beginning of training. The replay buffer is a priority queue indexed by the reward. Under-performing chains of thoughts are evicted as we collect more trajectories. We ablate the effect of seeding the replay buffer with inverse-prompted chains of thoughts and aggregating multiple chains at test time in Table D.4.

## E  INTEGER ARITHMETIC

**Additional details.**  See Table E.1 for examples from synthetically generated training data for the integer arithmetic task.

As with the infilling task, for the integer arithmetic task, the pretrained base models we tested were imperfect knowledge models, *i.e.* incorrect rationales are sometimes assigned very high likelihood. To assign the correct rationales higher rewards, we prepend some demonstrations with hand-crafted rationales to the query when computing the reward, *i.e.* $p_{\text{LM}}(XZY|(X_iZ_iY_i)_{i=1}^k)$ where $k = 3$ in our experiments. This improves the calibration to some extent. Note that these $(X_i, Z_i, Y_i)$ are taken from the dataset and used only in the reward. The GFlowNet policy here is conditioned only on $X$, *i.e.*, $q_{\text{GFN}}(Z|X)$.

We detail the hyperparameters used for training GFlowNets in our experiments in Table E.2. During training, we sample $(X, Y)$ from the dataset and then sample (batch size) $Z$s for every $(X, Y)$, before using $p_{\text{LM}}(XZY|(X_iZ_iY_i)_{i=1}^k)$ as the reward. During the generation of $Z$, as mentioned in §4.4, whenever a "=" is generated we extract the preceding expression and evaluate the last two terms using `eval` in Python. We use the replay buffer described in §3.3 and seed it with rationales from the dataset $p_{\text{LM}}$. We linearly anneal the reward temperature, and the learning rate during warmup. During evaluation, for all methods, we aggregate the response over multiple samples drawn from the model at some fixed temperature. For the zero-shot baseline, we observe the best performance with the "`Let us think step by step.`" prompt appended at the end of the question. For supervised fine-tuning and PPO, we use the implementation from `trl`. For supervised fine-tuning we use a batch size of 128 with 8 gradient accumulation steps and train for 10 epochs with a learning rate of 0.0001 using LoRA. For PPO we use a minibatch size of 64 with 16 gradient accumulation steps, a learning rate of 0.0001, 4 epochs (on the minibatch), a clip range of 0.2, and an adaptive KL coefficient initialized at 0.2.

**Effect of the number of rationales.**  As mentioned in §4.4, we seed the buffer with rationales from a dataset. Table E.3 summarizes the results of ablation on the number of examples used to seed the buffer. We observe, as expected, that the performance generally improves as we the number of examples used to seed the buffer grows. When no buffers are used to seed the buffer, the performance is quite poor. We hypothesize that without the good rationales used to seed the buffer, the exploration problem of searching for good rationales becomes very challenging.

Table C.3: Examples of infills generated by the prompting baseline along with **reference infills** for the stories infilling task.

| Beginning ($X$) | Middle ($Z$) | End ($Y$) |
|---|---|---|
| Randy had recently separated from his wife. He felt very lonely and sad all the time. He considered trying to patch things up with Vera, his ex-wife. | **He decided to get a puppy instead** He now happily had a new wife who he really liked. His wife missed him, but for the wrong reasons. Now He finally found someone he could be happy with. It was He began spending more time with his new girlfriend and friends. Randy remained single and mostly satisfied with that. He found his When Randy was happy he went to the aquarium, a place He started going out more with a display of male affection. He bought Vera a moonstone and gave it to her as He and Vera got back together, even though they had stayed He introduced Paige to his new employees and they were great. Randy is happily married to Vera and they have a healthy relationship He found a new lady with a lot of the same interests | His decision made him happy and he no longer felt sad. |
| Ken put a bottle of beer in the freezer. He heard a popping noise. He looked in the freezer and saw the bottle had burst. | **He didn't want to wait for another beer to get cold** He went to sleep and came awake with an angry stomach. Ken had another beer as a treat. After the third beer He stored the beer in the freezer and forgot about it. He walked around the bar a bit. He thinks he'll Weeks later, Ken preferred his cold beer. However, he He poured a fresh bottle with a sigh of relief. It He went to the bar for cold beer. No one knew Ken hung up the phone on his friend. He didn't He bought some veggies from the grocery store. It was warm Ken watched the movie Goalkeeper. He was glad the cold He laid down for a rest. His belly was full! He laid down for the night and finally sleep again. The | He drank a warm beer instead. |

Table C.4: Additional examples of infills generated by the prompting baseline along with **reference infills** for the stories infilling task.

| Beginning ($X$) | Middle ($Z$) | End ($Y$) |
|---|---|---|
| Jasmine had homework to do. She did not have a pencil. She looked around the room for one. | **The room did not have a pencil**
Jasmine made her pencil and done her work. She looked
Her mom brought her a pencil. Jasmine wrote her homework
Jasmine took her pencil out of the room. She now
Jasmine pretended not to have done her homework. She got
Her brother finally brought a pencil. They could finally start!
Her brother brought a pencil and solved her problem. Jasmine
Her brother brought a pencil and solved her problem. Jasmine
Her friend offered her a pencil. Jasmine used the pencil
Her teacher saw her looking at her phone. Then, Jas
Her mom came in and handed her the extra pencil! Jas
Her mom brought her a pencil. Jasmine then got to
Jasmine sent her friend money to do it. She was | Jasmine decided not to do her homework. |
| Jane had recently gotten a new job. She was nervous about her first day of work. On the first day of work, Jane overslept. | **Jane arrived at work an hour late**
Despite arriving late, Jane was greeted warmly by her coworkers
Jane learned that you never can predict the future. It was
Jane learned not to rush into anything. Jane made a good
Many of her peers were not pleased with her. Jane got
Jane ended up quitting her new job early. She was happy
More time with the wrong people gave her anxiety. She was
Jane learned that you do not mess up at a job well
Jane realized she needed to find new work contacts.'. At
After her first day of work, Jane was too tired to
Secretly, Jane hated her new job. It made her
Jane realized she worked at a mid-sized company with long
Jane's boss realized she did not show up to work on | Jane did not make a good impression at her new job. |

Table C.5: Examples of infills generated by the supervised fine-tuned model along with **reference infills** for the stories infilling task.

| Beginning ($X$) | Middle ($Z$) | End ($Y$) |
|---|---|---|
| Randy had recently separated from his wife. He felt very lonely and sad all the time. He considered trying to patch things up with Vera, his ex-wife. | **He decided to get a puppy instead**
He started enjoying his time with Vera even more. Now,
Eventually Randy missed Vera very much and developedWhile making memories with
He finally decided he had enough pain for no good reason.
He began spending more time with his new girlfriend. Now he
For the first time since the divorce he started to make new
Soon Randy and Vera were rekindling their romance. They
He started seeing someone new a day. Now, he feels
He signed up for a class in Same Sex Loving Uniqueness
For the first time in months, Randy and Vera celebrated a
He introduced Paige to his new employees. They are now good
After some contemplating, he decided to patch things up with Vera
He now looks for another partner to enjoy his life with. | His decision made him happy and he no longer felt sad. |
| Ken put a bottle of beer in the freezer. He heard a popping noise. He looked in the freezer and saw the bottle had burst. | **He didn't want to wait for another beer to get cold**
The popping noise was a freezer needle getting caught. It fell
Ken had a cold for thirty minutes. After the popping noise
He stored the beer in the freezer. The pouring sound was
He didn't know why it didn't stop the ringing.
Weeks later, Ken knew the reason for the freezing. It
Later Ken learned the gas is out of gas and will end
He left the freezer for the beer. It stayed in there
Before he knew it, the popping noise was gone. He
The popping noise that Ken heard indicated it should have freezer shut
The popping noise became louder and more frequent. Now Ken's
He froze the beer and drank it from a mason jar
He went to the store to buy another. Now, he | He drank a warm beer instead. |

Table C.6: Additional examples of infills generated by the supervised fine-tuned model along with **reference infills** for the stories infilling task.

| Beginning ($X$) | Middle ($Z$) | End ($Y$) |
|---|---|---|
| Jasmine had homework to do. She did not have a pencil. She looked around the room for one. | **The room did not have a pencil**
Jasmine decided to talk to her mom about it. They
Her dad now has to teach her math! Picking up
Jasmine took her math test later in the day. She
Jasmine got an A on her assignment. She continued to
Her brother finally brought a pencil. They could finally start!
Her brother brought a pencil and solved her problem. Jasmine
Jasmine used her pink and brown highlighter instead.
Jasmine went to school. She was able to finish her
Jasmine made her own pencil. At school, everyone was
Her brotherly earrings helped her stay up more. While
Jasmine made her own pencil. She ended up getting good
Jasmine sent her sister to do it. Now it was | Jasmine decided not to do her homework. |
| Jane had recently gotten a new job. She was nervous about her first day of work. On the first day of work, Jane overslept. | **Jane arrived at work an hour late**
After arriving home, Jane realized her apartment was not ready.
After work, Jane returned home and ate a lousy sandwich.
After her first day of work, Jane learned that mistakes happen
Due to her bad first day, Jane was fired from her
Ending: Jane ended up late for work. Although she
On her 2nd day of work, Jane came home late
On the second day of work, Jane arrived late and unprepared
By the end of the day, Jane was tired and weary
After her first day of work, Jane's boss recommended her
Secretly, Jane hated her new job. Secretly,
After work, Jane felt worn out and tired. The end
By the end of the day Jane felt confident and like she | Jane did not make a good impression at her new job. |

Table C.7: Examples of infills generated by the GFlowNet fine-tuned model along with **reference infills** for the stories infilling task.

| Beginning ($X$) | Middle ($Z$) | End ($Y$) |
|---|---|---|
| Randy had recently separated from his wife. He felt very lonely and sad all the time. He considered trying to patch things up with Vera, his ex-wife. | **He decided to get a puppy instead**
He went to a friend's house to talk.
He took Vera to the park for a picnic.
He finally decided to call Vera on the phone.
He asked Vera to meet him and they did.
He decided to get a dog for his family.
He went to Vera's house.
He told Vera that he was ready to love again.
He bought Vera a nice gift.
He had two dogs and felt like he missed Vera.
He had a good couple of years.
He wanted to get Vera and he eventually did.
He wanted to sit in a room alone. | His decision made him happy and he no longer felt sad. |
| Ken put a bottle of beer in the freezer. He heard a popping noise. He looked in the freezer and saw the bottle had burst. | **He didn't want to wait for another beer to get cold**
He decided to stop drinking beer.
Ken suddenly smelled a bottle of beer.
He thought it was something
He walked around the freezer to see the bottle.
He tried to clean up the freezer.
He was angry when he had to drink it.
He stepped down from the freezer
He made a big joke that night.
He looked in a jug and see it was, out.
He then looked for a beer the freezer.
He tried to enjoy the rest.
He watered the bottle, | He drank a warm beer instead. |

Table C.8: Additional examples of infills generated by the GFlowNet fine-tuned model along with **reference infills** for the stories infilling task.

| Beginning ($X$) | Middle ($Z$) | End ($Y$) |
|---|---|---|
| Jasmine had homework to do. She did not have a pencil. She looked around the room for one. | **The room did not have a pencil**
She was in a panic.
She had to place the pencil in her pocket.
She had several her classmates.
She looked for a neat pencil.
She thought she looked everywhere.
Her brother walked in and borrowed her pencil.
She only had a few minutes to do it.
She never had her pencil.
She saw a pen in her closet.
She searched everywhere for a pencil.
She went to her room.
She didn't find one. | Jasmine decided not to do her homework. |
| Jane had recently gotten a new job. She was nervous about her first day of work. On the first day of work, Jane overslept. | **Jane arrived at work an hour late**
She lost her job after the first day of work.
She was egged on by her boss.
She decided to over hyp her first day.
She arrived late and missed her first day of work.
She was nervous about going to work.
She ended up getting a good job.
Jane waited almost a day to get to work.
Her first day was very difficult.
She made a couple of phone calls.
She arrived at the office and was fired.
She was all out of coffee.
Jane was surprised she did not make a good impression. | Jane did not make a good impression at her new job. |

Table D.1: Two training examples from the SUBJ dataset.

| Text ($X$) | Label ($Y$) |
|---|---|
| another story follows the relationship between a stepfather ( neeson ) and his young stepson . | objective |
| hoffman 's performance is authentic to the core of his being . | subjective |

Table D.2: Prompts used for subjectivity classification.

| Few-shot learning / Supervised fine-tuning | GFlowNet fine-tuning |
|---|---|
| Classify this movie review as objective or subjective: "[X]" This review is [Y]. | Classify this movie review as objective or subjective: "[X]" This review is [Z], so it is [Y]. |

Table D.3: Hyperparameters for GFlowNet fine-tuning on subjectivity classification.

| | |
|---|---|
| LoRA rank | 256 |
| LoRA scaling factor | 16 |
| LoRA dropout | 0. |
| Batch size | 16 |
| Gradient accumulation steps | 32 |
| Learning rate | 0.0005 |
| Optimizer | AdamW |
| $P_F$ temperature max | 2 |
| $P_F$ temperature min | 0.5 |
| Reward temperature start | 1.2 |
| Reward temperature end | 1.0 |
| Reward temperature horizon | 150 |
| Buffer capacity | 50 |
| Number of steps | 100 |
| Number of samples | 10 |
| Maximum length | 5 |
| Minimum length | 1 |

**Analysis.** In Table E.4 we show some examples generated by PPO and GFlowNet fine-tuned models. We observe that PPO generates sequences with high rewards under the base model. These sequences, however, are not valid expressions, and, in fact, do not even call the tool. Instead, the model learns to simply repeat the expression. Repetitions being assigned high likelihood is an issue that has also been noted by Welleck et al. (2020). On the other hand, GFlowNets are able to generate the correct rationale to evaluate the expression.

There are still cases where the GFlowNet fails to produce the correct reasoning chain. We illustrate some of these examples in Table E.5. The errors fall mainly into 3 categories: 1) missing operands in longer OOD problems 2) incorrect operand being copied 3) incorrect operator being copied. The potential source for such errors is that the reward model assigns equally high rewards to rationales with these minor mistakes and thus the model generates them with likelihood. These errors can potentially be reduced by using better and potentially larger reward models.

Table D.4: Ablation studies on subjectivity classification using GPT-J 6.8B.

| Method | Test accuracy (%) ↑ | | |
|---|---|---|---|
| | Training samples | | |
| | 10 | 20 | 50 |
| GFlowNet fine-tuning | 71.4 ± 1.8 | 81.1 ± 0.4 | 87.7 ± 2.2 |
| (-) Seed buffer | 64.7 ± 8.9 | 68.7 ± 1.7 | 77.0 ± 5.5 |
| (-) Seed buffer (-) Aggregating | 63.9 ± 7.1 | 65.9 ± 2.4 | 75.4 ± 3.3 |

Table D.5: Top sample rationales for the SUBJ test set using the instruct-fine-tuned GPT-J 6B as both the reward model and the base model for the GFlowNet, which is trained on 50 labeled examples.

| True label | Rationale | | Frequency |
|---|---|---|---|
| Objective | This review is | about a factual event | 11.45% |
| | | describing a factual event | 10.23% |
| | | based on a factual statement | 9.47% |
| | | about a historical event | 8.77% |
| | | about a movie review | 7.41% |
| | | based on facts | 5.64% |
| | | about a factual statement | 4.61% |
| Subjective | This review is | about a movie review | 26.21% |
| | | about the movie experience | 26.17% |
| | | about the movie | 7.60% |
| | | describing a movie review | 3.44% |
| | | about the movie review | 2.31% |
| | | based on a fictional story | 2.30% |
| | | describing the movie experience | 2.01% |

Table E.1: Examples from the integer arithmetic dataset. Note that the results of the two-term expressions are evaluated by the calculator.

| **Question** ($X$) | **Rationale** ($Z$) | **Answer** ($Y$) |
|---|---|---|
| Question: 6 - 0 - 4 - 8 = ? Answer: | 6 - 0 =, 6 - 4 =, 2 - 8 = | . The answer is -6. |
| Question: 9 + 4 - 8 = ? Answer: | 9 + 4 =, 13 - 8 = | . The answer is 5. |

Table E.2: Hyperparameters for the Integer Arithmetic Task

| | |
|---|---|
| LoRA rank | 64 |
| LoRA scaling factor | 16 |
| LoRA dropout | 0.1 |
| Batch size | 16 |
| Gradient accumulation steps | 32 |
| Learning rate | 0.0001 |
| Optimizer | AdamW |
| $P_F$ temperature max | 2 |
| $P_F$ temperature min | 0.5 |
| Reward temperature start | 1.1 |
| Reward temperature end | 0.5 |
| Reward Temperature horizon | 150 |
| Buffer capacity | 50 |
| Number of training steps | 1000 |
| Evaluation temperature | 0.1 |
| Number of samples | 10 |
| Maximum length | 20 |
| Minimum length | 5 |

Table E.3: Ablation study to understand the effect of the number of rationales used to seed the buffer on the test accuracy (%) for the integer arithmetic task.

| | Number of operands | | |
|---|---|---|---|
| | In-distribution | | OOD |
| Number of seed rationales | 3 | 4 | 5 |
| 0 | 22.6 | 5.8 | 3.4 |
| 10 | 58.6 | 52.3 | 20.2 |
| 25 | 48.8 | 56.8 | 24.6 |
| 50 | 95.2 | 75.4 | 40.7 |

Table E.4: Samples generated by PPO fine-tuned and GFlowNet fine-tuned models.

| Question ($X$) | Generated rationale ($Z$) | $\log R$ |
|---|---|---|
| Question: 1 - 9 + 8 = ? Answer: | 1 - 9 - 8 | -13.17 |
| | 1 - 9 = -8, -8 + 8 = 0 | -27.75 |
| Question: 8 + 7 + 2 + 7 = ? Answer: | 8 + 7 + 2 + 7 | -2.39 |
| | 8 + 7 = 15, 15 + 2 = 17, 17 + 7 = 24 | -11.72 |
| Question: 7 - 5 + 8 - 0 - 6 =? Answer: | 7 - 5 + | -1.22 |
| | 7 - 5 = 2, 2 + 8 = 10, 10 - 0 = 10, 10 - 6 = 4 | -7.99 |

Table E.5: Examples of errors made by the GFlowNet-fine-tuned model.

| Question ($X$) | Generated rationale ($Z$) |
|---|---|
| Question: 7 + 3 + 4 - 9 + 9 =? Answer: | 7 + 3 = 10, 10 - 9 = 1, 1 + 4 = 5 |
| Question: 8 + 5 - 1 - 4 + 5 =? Answer: | 8 + 5 = 13, 13 - 1 = 12, 12 + 4 = 16, 16 - 4 = 12 |
| Question: 0 + 5 - 6 - 1 - 6 =? Answer: | 0 + 5 = 5, 5 - 6 = -1, -1 - 6 = -7, -7 + 1 = -6 |

