# OpenReview forum: "Amortizing intractable inference in large language models"
_ICLR.cc/2024/Conference — ICLR 2024 oral_

### Official Review · Reviewer_7WZA · 2023-10-30

**Soundness:** 4 excellent
**Presentation:** 4 excellent
**Contribution:** 4 excellent
**Rating:** 8
**Confidence:** 3

**Summary:**

LLMs are good at auto-regressive sampling, as that is how they are defined and trained.  However, there are many types of inference queries can make LLMs more useful by allowing involved reasoning.  There are multiple examples where this involves sampling from intractable posterior distributions.

GFlownets (GFN) provide a way to fine-tune LLMs for more specific inference queries as it can tune the generative distribution to a non-next-token reward function.  This is particularly relevant as many current reasoning methods (e.g., chain of thoughts), can instead be thought of as alternative inference queries in a probabilistic model.

To motivate the method, the paper illustrates that GFN, in contrast to PPO, can match a posterior distribution over random numbers, while PPO can only ensure that all the samples are valid, without ensuring that the distribution can match.  Supervised fine-turning also works here, but not when there are no samples available to match with, which is not always the case.

The paper then goes over a variety of interesting intractable distributions that can now be sampled from with GFNs, allowing tasks like non-local low temperature sampling, infilling and constrained generation.  Note that each of these tasks requires separate and inference specific fine-tuning.  So while GFNs can sample high quality and diverse low temperature sentences by fine tuning for a given temperature, diverse beam search requires 5x more compute at inference time, but also doesn’t require retraining.

The paper also shows how to use variational EM to optimize the chain of thought reasoning to get the correct reasoning without providing additional training data for the reasoning.

The paper then demonstrates the empirical benefits of their approach in 4 experiments, low temperature sampling, infilling, subjectivity classification and solving arithmetic problems, in each case demonstrating their superior performance to good baselines.

**Strengths:**

## Originality:
The paper seems to be a useful and novel contribution to the literature, namely the using GFNs to fine tune LLMs to solve formley intractable inference problems inside LLMs.  While the learning objective itself has been suggested before for LLMs, there does not appear to be any follow up work in applications like the low temperature sampling, infilling and learning chains of reasoning as in the current work.  It also appears to be a novel method for training chain of thought reasoning to arrive at a specific outcome.

## Quality:
The authors clearly reference the contemporary literature and they compare against suitable baselines in their experiments.  Overall, they have clearly demonstrated a variety of strong results against suitable baselines.

## Clarity:
The paper is clearly written and structured in an easy to follow manner.  The descriptions of the methods and experimental setups are complete enough that the results can easily be reproduced by other interested parties.

## Significance:
LLMs are intrinsically highly significant at the moment and contain a huge amount of relevant information about the world which can't always reliably be extract out, so better methods to run interesting queries on them will have a practical effect. After all, they demonstrate that by this fine-tuning approach they can extract more valuable information from the same LLM, requiring only (presumably) extra inference time work not more data.   On the more theoretical level they demonstrate that GFNs can useful scale to large models, motivating further explorations of such methods in the current era of large models.

Also, in particular, learning better chains of reasoning could in particular have many interesting planning and reasoning applications, which could be unlocked with future research.

**Weaknesses:**

The datasets are small, and the inference only involved fine-tuning vs. training from scratch, which may unlock entirely different and interesting new global solutions.  This is an understandable limitation, but a more full exploration (which can be tackled as future work) might extend the power and reach of their method.

One of the central pieces, the learning objective, has already been derived in a different soft-RL Q learning context, as pointed out by the authors.  There doesn’t appear to have been further explorations of downstream applications as performed in the current work however.

Their method requires fine-tuning for different queries/inference problems.  It would be interesting to see if instead it would be possible to have a single network (as they suggest in future work) that can answer many different types of queries.

**Questions:**

What do the reference in-fills look like for the other distributions?  It only shows the GFlowNet examples in Table B.3.

What’s the speed of learning/convergence relative to, e.g. supervised fine tuning?  Is the inference unstable/need many restarts, etc?  How about compared to PPO training?

Did the authors fine-tune separately for every temperature, or was this done in the style of an amortized sampler where you can dynamically specify the target temperature at run time?

---

> ### Author Response · Authors · 2023-11-17
> **Thank you for your review!**
>
> We thank the reviewer for their helpful comments on our work.
>
> > What do the reference in-fills look like for the other distributions? It only shows the GFlowNet examples in Table B.3.
>
> We have added samples generated by each method on 4 randomly selected examples to Appendix C in the updated manuscript.
>
> > What’s the speed of learning/convergence relative to, e.g. supervised fine tuning? Is the inference unstable/need many restarts, etc? How about compared to PPO training?
>
> In terms of wall time, GFlowNet fine-tuning is slower compared to supervised fine-tuning as it has to explore the space of sequences and has a similar runtime to PPO. We did not encounter instabilities during training that required special treatment like restarts, etc.
>
> > Did the authors fine-tune separately for every temperature, or was this done in the style of an amortized sampler where you can dynamically specify the target temperature at run time?
>
> We do not amortize over the temperature, though it has been studied in some prior work on GFlowNets. We linearly anneal the reward over the course of training.
>
> We hope we have addressed all of your concerns. Please don't hesitate to let us know if we can clarify anything else!

---

> > ### Comment · Reviewer_7WZA · 2023-11-20
> >
> > Thanks for your answers to my questions, the extra details about the training and convergence behavior and the reference sampels.
> >
> > One very minor point in the response - please don't highlight text in red, as it is almost impossible to spot vs. black text if you're red-green color blind.

---

> > > ### Author Response · Authors · 2023-11-20
> > > **Changed colours**
> > >
> > > Thanks for the suggestion. We have uploaded a revised pdf with the corrections marked in blue. To distinguish them from hyperlinks (such as those in citations), we have made the links a bright green colour. In the final version, we will remove the highlighting of changes and make the links blue again.

---

### Official Review · Reviewer_Vqdw · 2023-11-01

**Soundness:** 3 good
**Presentation:** 3 good
**Contribution:** 3 good
**Rating:** 8
**Confidence:** 4

**Summary:**

This paper investigates the challenge of sampling latent variables from a posterior distribution in large language models (LLMs), where the latent variables might take the form of prompts, reasoning chains, etc. However, sampling from a posterior distribution is typically intractable. To address it, the paper proposes to use generative flow networks (GFlowNets), which sample a composite object (a sequence of tokens) via a sequence of constructive steps, with a probability proportional to a reward function (the product of the likelihood and the prior, leading to the joint distribution). This is different from MLE-based fine-tuning and reward-maximization-based fine-tuning, which tend to make the learned distribution more concentrated on one or few modes, potentially leading to incorrect outputs. In contrast, Bayesian inference aims to learn a distribution that encompasses all possible outputs, thus promoting diversity and preventing from overfitting to a wrong target. The authors used a modifed version of the SubTB training objective for fine-tuning and their experimental results demonstrate the effectiveness of GFlowNets-based fine-tuning in improving text generation and reasoning tasks.

**Strengths:**

### Motivation:

One limitation of existing fine-tuning techniques, i.e., MLE and reward maximization - the learned distribution will end up focusing around one or very few outputs, due to the nature of maximization. If the wrong one was picked up, the consequence could be catastrophic. This is where the Bayesian posterior comes in - it can contain all the information over the potential outputs. However, sampling from a posterior distribution is typically intractable. GFlowNets have recently been shown to approximate a complicated multimodal distribution well. To this end, GFlowNets are used to sample composite latent variables via a sequence of steps, with a probability proportional to $p_{LM}(XZY)$ or $p_{LM}(XZ)$.


### Originality:

The proposed GFlowNet fine-tuning builds on GFlowNets and Bayesian posteriors. The authors utilize GFlowNets as an amortized inference machine, to sample composite latent variables from an intractable posterior distribution in LLMs. This is different from MLE-based and reward-maximization-based fine-tuning techniques. The resulting GFlowNet fine-tuning shows good performance on various tasks. The originality is good.


### Clarity:

The paper is well-organized.

**Weaknesses:**

Please see the following questions.

**Questions:**

## Amortized inference with GFlowNet Objectives

- GFlowNets start with an empty string and add one token at a time in a left-to-right manner. Depending on different tasks, $Z$ should be generated conditional on $X$ or $X, Y$? Here, $X$ or $X, Y$ is omitted?

### Learning objective

- Besides that 1) the ability to avoid to estimate the flow function $F$; 2) SubTB can have a better bias-variance trade-off in GFlowNet training, are there any other benefits to use a modified version? Also, did you try the conventional SubTB objective with the flow function considered?

- Did you consider the hyper-parameter $\lambda^{j-i}$ over incomplete trajectories with variable lengths $0 \leq i < j \leq n+1$, like SubTB($\lambda$)?

- Given that the generation order is fixed (i.e., left-to-right), it results in $P_{B} = 1$. For readers who might be unfamiliar with GFlowNets, it might be helpful to include an explanation or mention $P_B=1$ somewhere in the paper to ensure accessibility for all readers?

### Parameterization, amortization, and generalization

- $R(Z) = p_{LM}(XZY) \propto p_{LM}(Z | X, Y)$ --> should be $p_{LM}(Z | X, Y) \propto R(Z) = p_{LM}(XZY)$?


## Empirical results

- How to understand GFlowNet fine-tuning and supervised fine-tuning solely? The former is to train the LM with Eq.3, while the later is to train the LM by maximizing $\log p_{LM}(XZY)$ with $Z$. Supervised fine-tuning corresponds to the variational EM - first update GFlowNet polices and then LM pamemters with $Z$? Thus, supervised fine-tuning should already include GFlowNet fine-tuning?

- In Table 3, GFlowNet fine-tuning + supervised fine-tuning was considered. Then why not to consider it as well in Table 2 & 4?

### 4.1 Sentence continuation - task description

- $R(Z) = p_{LM}(Z | X)^{\frac{1}{T}}$ --> should be $R(Z) = p_{LM}(XZ)^{\frac{1}{T}}$?

---

> ### Author Response · Authors · 2023-11-18
> **Thank you for your review!**
>
> We thank the reviewer for their helpful comments on our work.
>
> > GFlowNets start with an empty string and add one token at a time in a left-to-right manner. Depending on different tasks, $Z$ should be generated conditional on $X$ or $X, Y$? Here, $X$ or $X, Y$ is omitted?
>
> In the subjectivity and arithmetic experiments, the GFlowNet depends only on $X$ so we can use it on unseen problems at test time when $Y$ is not available.
> $Y$ is available at test time for the infilling experiment, so we the GFlowNet is conditioned on both $X$ and $Y$.
>
>
> > Besides that 1) the ability to avoid estimating the flow function $F$; 2) SubTB can have a better bias-variance trade-off in GFlowNet training, are there any other benefits to use a modified version? Also, did you try the conventional SubTB objective with the flow function considered?
>
> The primary motivation for the modification in our approach is to avoid learning a state flow function, limiting the learnable objects to just the forward policy (i.e., the same data that is output by an autoregressive LM). We did not experiment with the standard SubTB objectives since it would add additional complexity (adding an extra head to the LM to output the flow) not central to our main contribution.
>
> > Did you consider the hyper-parameter $\lambda^{j-i}$ over incomplete trajectories with variable lengths $0 \leq i < j \leq n+1$, like SubTB($\lambda$)?
>
> We did not tune the SubTB hyperparamter $\lambda$ and left it to the default value of 1 for all experiments following prior work on GFlowNets, such as [Hu et al., 2023].
>
> > Given that the generation order is fixed (i.e., left-to-right), it results in $P_{B} = 1$. For readers who might be unfamiliar with GFlowNets, it might be helpful to include an explanation or mention $P_B=1$ somewhere in the paper to ensure accessibility for all readers?
>
> Thank you for the feedback! We have added a note to clarify this in the updated draft.
>
> > $R(Z) = p_{LM}(XZY) \propto p_{LM}(Z | X, Y)$ --> should be $p_{LM}(Z | X, Y) \propto R(Z) = p_{LM}(XZY)$?
>
> We may be misunderstanding your comment: those two are equivalent, if we are talking about proportionality in $Z$. The expression is intended to define the reward for $Z$ as the likelihood $p_{\text{LM}}(XZY)$ which is the quantity proportional to the desired posterior. Thus we state it as $R(Z) = p_{\text{LM}}(XZY) \propto p_{\text{LM}}(Z | X, Y)$.
>
> > How to understand GFlowNet fine-tuning and supervised fine-tuning solely? The former is to train the LM with Eq.3, while the later is to train the LM by maximizing $\log p_{LM}(XZY)$ with $Z$. Supervised fine-tuning corresponds to the variational EM - first update GFlowNet polices and then LM pamemters with $Z$? Thus, supervised fine-tuning should already include GFlowNet fine-tuning?
>
> GFlowNet fine-tuning corresponds to only the E-step in EM. Supervised fine-tuning on its own directly maximizes $\log p_{LM}(Y|X)$ without a latent variable $Z$.
>
> The supervised fine-tuning on top of GFlowNet fine-tuning, however, maximizes $\log p_{LM}(Y|XZ)$ with $Z$ drawn from the GFlowNet (this is the M-step in EM).
>
> > In Table 3, GFlowNet fine-tuning + supervised fine-tuning was considered. Then why not to consider it as well in Table 2 & 4?
>
> In story infilling (Table 2), the goal is to generate $Z$ given both $X$ and $Y$, so there is no need to finetune $p_{LM}(Y|X, Z)$. For the tool use problem (Table 4), the $Z$ already contains the solution and consequently, there isn't much improvement to be expected with subsequent supervised fine-tuning of the LM. Thus, we do not consider GFlowNet fine-tuning + supervised fine-tuning in those experiments.
>
> > 4.1 Sentence continuation - task description
> $R(Z) = p_{LM}(Z | X)^{\frac{1}{T}}$ --> should be $R(Z) = p_{LM}(XZ)^{\frac{1}{T}}$?
>
> These two rewards are equal up to a multiplicative constant that would only depend on $X$, i.e., $p_{LM}(XZ)^{\frac{1}{T}} = p_{LM}(Z|X)^{\frac{1}{T}}p_{LM}(X)^{\frac{1}{T}}$. The conditioning variable $X$ is given as input to the GFlowNet policy. This means that both rewards describe the same posterior distribution. Therefore, in theory, an optimally-trained GFlowNet would converge to the same solution.
>
> However, in practice, using $R(Z) = p_{LM}(XZ)^{\frac{1}{T}}$ could be more difficult to optimize because the scale of the reward could differ significantly for different $X$'s (e.g., sentences following very long prompts would generally have much lower rewards than sentences following shorter prompts). The choice of using $R(Z) = p_{LM}(Z | X)^{\frac{1}{T}}$ corresponds to substracting a baseline from the log-reward that depends only on $X$, a common variance reduction technique in RL.

---

> > ### Comment · Reviewer_Vqdw · 2023-11-20
> > **Thank you for your response!**
> >
> > Thank you for your detailed response. They are clear to me. I appreciate your efforts to improve the paper.

---

### Official Review · Reviewer_H3x9 · 2023-11-02

**Soundness:** 4 excellent
**Presentation:** 3 good
**Contribution:** 4 excellent
**Rating:** 10
**Confidence:** 4

**Summary:**

This paper presents a new technique for fine-tuning LLMs, to perform amortized inference in probabilistic models (also defined using LLMs). This enables tuning LLMs for more interesting objectives than traditional RL or supervised fine-tuning techniques. The authors present several examples of such objectives: optimizing chain-of-thought reasoning so that it more often leads to the correct answer, optimizing for useful tool use, infilling plausible middles of stories with beginnings and ends, and whole-sentence temperature sampling for higher-quality sentence completion. In each, the proposed method is shown to outperform baselines.

**Strengths:**

This is a very strong paper. Some of its key strengths are:

* There is a very nice, pedagogical discussion of why it may be desirable to sample intractable distributions, which provides great motivation for the proposed approach. The random-number example in Section 2 also nicely illustrates the limitations of reinforcement learning.

* Several researchers have proposed using "online" (i.e., test-time) inference methods to sample LLM posteriors, but those methods are only appropriate in settings where the increased cost of exploring multiple samples at test time is not prohibitive. This paper's technique enables offline training, and produces a network that can generate approximate posterior samples directly at test time. What's more, even in settings where test-time inference *is* feasible, Monte Carlo algorithms could use the amortized networks introduced by this paper as proposals, to rapidly speed convergence in cases the amortized networks handle well, and more gracefully handle cases where the amortized networks do not generalize.

* Unlike some formulations of amortized inference, which require exact posterior samples to fine-tune on (e.g., https://arxiv.org/abs/1610.05735), this paper requires only the ability to evaluate the unnormalized posterior (the reward $R$).

* The experiments suggest that this technique is applicable to a compellingly broad range of tasks. The experiments showing that the technique can help train LLMs to perform better reasoning over latent variables (the thoughts in chain-of-thought, or the tool invocations in tool-use applications) are particularly nice.

* The writing is clear (if somewhat terse) throughout.

**Weaknesses:**

Overall, I really like the paper, but I do think there are a few places it could be improved:

1. **Limited discussion of the training objective and its relationship to possible alternatives.** The training objective is introduced very briefly and without much intuition. I realize that there is an extensive literature on training GFlowNets and there is not space to go into full detail here. But are there reasons that this objective (among many other GFlowNet objectives) was particularly well-suited to the language modeling case? Why GFlowNets at all instead of e.g. reweighted wake sleep (a common method for amortizing intractable posterior inference)? How sensitive is performance to the distribution you use to generate training trajectories? How important is the replay buffer, and how is it populated? In fairness, I am not sure how many of these questions need to be addressed in a short conference paper.

2. **Limited discussion of the limitations of the proposed technique.** Ultimately, the training method given here is a mostly-on-policy reinforcement learning method. A key challenge for such methods is exploration -- finding high-reward samples to reinforce. I would have appreciated more discussion of the sorts of posterior inference tasks that are and aren't likely solvable with the proposed techniques (at least without further innovations), possibly along with potential mitigations for these weaknesses.

3. **Metrics for infilling.** I had reservations about some of the metrics used to evaluate the proposed approach, in particular for the story infilling task. It is unclear that measuring similarity to a single reference sentence is very meaningful--especially since a purported strength of the method is sampling the full posterior. It would be nice if (randomly selected) qualitative examples were presented for all baselines. It may also be worth considering an automated evaluation of the coherence of the resulting story (e.g., by asking GPT-4 to rate coherence). Despite the many (valid) critiques of such LLM-powered evaluations, I do think they are at least a better fit for creative coherent generation tasks like this one than metrics like BLEU.

**Questions:**

# Questions

* Around how long (in wall-clock time) does it take to LoRA fine-tune a GFlowNet on your tasks, e.g. for a 6B-parameter model?

* In Table 3, how were Test Accuracy numbers in the final row ("+ Supervised Fine-Tuning") generated? Were 10 samples of Z taken from the fine-tuned model, and aggregated via voting? Or are the Z samples still generated from q_{GFN} but now completed with Y drawn from the fine-tuned LM? Or is there no longer a voting procedure?

* In principle, for tasks like story infilling, supervised fine-tuning (SFT) should be optimizing the same architecture as the GFN for an objective that has the same optimum (i.e., SFT is also a distribution-matching objective, where the optimum is the intractable posterior). Qualitatively, how do the samples from the SFT baseline look? (I think it would be nice to add them to Table B3 if possible!) If they are noticeably worse than the GFN samples, what would you attribute that to? Also: for SFT and for the "just prompt the model to infill" baseline, do you start with the base language model, or the reward language model that you fine-tuned with stories?

* What is "reward temperature horizon"? What are the P_F min and max temperatures? (I saw that reward temperature was annealed during training, but did not see a reference to annealing the QFN's own temperature.)

* You write on p23 that the reward model could often not distinguish between good and bad rationales. Does this mean that, given a prompt (e.g.) Z="..., 1 + 4 = 5. The answer is:", the reward model assigns roughly equal probability to (a) the known correct answer Y from the training data (e.g., Y=14) and (b) the most-recently computed number (in this case, 5)? That's somewhat surprising to me!

* For sentence continuation, do you see interesting pathologies at lower reward-model temperatures (e.g., bias toward very short completions, or very repetitive completions)?

# Minor Comments

* There are a couple points that I found confusing when first reading the paper, even though they are clarified later.

  (1) The clause "finding the most likely sequence continuation" in the first paragraph was confusing. "Finding likely sequence continuations" is precisely what LLMs are trained to do, and would not seem to require intractable inference; I considered briefly that you might mean finding the literal maximum-probability sequence, but that also seemed wrong because I was expecting a list of posterior sampling tasks, not optimization tasks. Later I realized that you meant long-range (i.e., not per-token) reduced-temperature sampling, but this wasn't obvious from the intro.

  (2) At multiple points you discuss chain-of-thought reasoning as an instance of intractable inference, with the formula P(Z | X, Y). But at test time, in chain-of-thought reasoning tasks, we do not *see* the answer Y, so it's not really a posterior sampling task. (If I give you only a single instance of a 'problem' in the chain-of-thought reasoning task, there is no clear MCMC target distribution you could specify over "good chains of thought" without already having access to the final answer Y. This is in contrast to the other tasks, like long-range temperature sampling, infilling, etc. where the reward $R$ can be evaluated at test time.)  After reading the whole paper, I have a clearer understanding of what it is you're doing in these (very neat) chain-of-thought examples. But their inclusion at the beginning of the paper, without sufficient explanation of how they work, makes it trickier to understand the proposed framework. Even later in the paper, there are two separate resolutions to the question of "what to do without Y" -- one is to use the (X, Y) pairs you have in order to generate Z's for fine-tuning (the "EM" idea), and the other is to train the GFN without access to Y, which could perhaps be interpreted as training it to do posterior inference conditioned on the event that the final answer is correct, rather than on a particular final answer.

* The idea that fine-tuning to do better chain-of-thought reasoning might be viewed as a kind of EM was previously proposed by Dohan et al. (although not implemented).

* At the bottom of p1, one of your citations is to a method that uses SMC, not MCMC, for which the notion of "mixing between modes" is not quite appropriate.

---

> ### Author Response · Authors · 2023-11-17
> **Thank you for your review!**
>
> We thank the reviewer for the strongly positive assessment and insightful comments on our work.
>
> > Limited discussion of the training objective and its relationship to possible alternatives. The training objective is introduced very briefly and without much intuition. I realize that there is extensive literature on training GFlowNets and there is not enough space to go into full detail here. But are there reasons that this objective (among many other GFlowNet objectives) was particularly well-suited to the language modeling case?
>
> We use the subtrajectory balance loss as it has favorable properties such as low gradient variance which enables stable training [Madan et al., 2023]. The modification to account for each state being a valid terminal state comes from [Deleu et al., 2022] and consists in replacing the flow function $F(z_{1:n})$ by $R(z_{1:n}\top)/q_{\text{policy}}(\top\mid z_{1:n})$. We make this choice because it allows us to instantiate the GFlowNet without explicitly parameterizing the flow function, only the policy itself.
>
> > Why GFlowNets at all instead of e.g. reweighted wake-sleep (a common method for amortizing intractable posterior inference)?
>
> GFlowNets have an advantage over other variational approaches -- the ability to train on off-policy trajectories without resorting to importance sampling. Approaches such as reweighted wake-sleep rely on importance sampling to utilize off-policy trajectories, which can lead to high-variance and biased estimates of the gradient. This finding was established by [Malkin et al., 2023].
>
> > How sensitive is performance to the distribution you use to generate training trajectories?
>
> The distribution of training trajectories we used is a mixture of on-policy trajectories, trajectories from a tempered policy, and trajectories from the replay buffer. The performance indeed depends on the choice of this distribution. Due to the compute constraints, we prioritized ablating factors such as seeding the replay buffer with good rationales over this aspect.
>
> > How important is the replay buffer, and how is it populated? In fairness, I am not sure how many of these questions need to be addressed in a short conference paper.
>
> The replay buffer is populated using trajectories sampled from the policy (and a tempered version of it) during training and some seed rationales at the beginning. Trajectories are added to the buffer based on a diversity threshold, i.e., a trajectory is added to the buffer only if it is a distance $\delta$ from every example in the buffer or it has a higher reward than the closest element, and the buffer is instantiated as a priority queue. Our analysis in Table D.4 and Table E.3 illustrates the importance of the number of examples used to seed the buffer. Additionally, we observe that the off-policy trajectories provided by the buffer are critical for reliable training.
>
> > For sentence continuation, do you see interesting pathologies at lower reward-model temperatures (e.g., bias toward very short completions, or very repetitive completions)?
>
> Indeed, we do see pathologies at lower reward temperatures, which is why the lowest temperature we provided results for was $T = 0.8$. For instance, when we train with $T = 0.6$ the average sentence length becomes only $1.58$ tokens, with a large proportion of sentences consisting of just an empty space followed by a period, or other short, common, and generic phrases (e.g., "We know."). Furthermore, the diversity decreases to $\sim 0.4$, compared to $\sim 0.75$ at $T = 0.8$. Despite this, the model samples sentences with high log-likelihood values that are comparable to what is achieved at higher temperatures ($-9.18$ maximum log-likelihood). This means that the pathologies are due to a combination of two factors: (1) the LM used for the reward has a bias for short and generic sentences, as is expected, and (2) training at lower temperatures harms exploration, as evidenced by the lower diversity and comparable log-likelihood of the samples. While the second problem can potentially be addressed with small modifications to our training scheme (e.g., more gradual reward temperature annealing), the first problem suggests that LM sentence log-likelihood on its own is a suboptimal reward signal for generating naturalistic sentences and that alternatives should be explored.

---

> > ### Author Response · Authors · 2023-11-17
> >
> > > Metrics for infilling. I had reservations about some of the metrics used to evaluate the proposed approach, in particular for the story-infilling task. It is unclear that measuring similarity to a single reference sentence is very meaningful--especially since a purported strength of the method is sampling the full posterior. It would be nice if (randomly selected) qualitative examples were presented for all baselines. It may also be worth considering an automated evaluation of the coherence of the resulting story (e.g., by asking GPT-4 to rate coherence). Despite the many (valid) critiques of such LLM-powered evaluations, I do think they are at least a better fit for creative coherent generation tasks like this one than metrics like BLEU.
> >
> > For the infilling task, we fill in the fourth sentence in the five-sentence stories from the RealStories dataset. This sentence typically involves some sort of "plot twist" which makes the beginning consistent with the end. By measuring the similarity of samples from the model with the reference we aim to judge how well this "plot twist" is captured by the model outputs.
> >
> > The BERTScore, GLEU and BLEU metrics with respect to reference responses have been used in prior work extensively, and the practice of measuring generated text quality with a combination of similarity to a reference and diversity metrics is common in areas such as dialogue response generation (see, e.g., [Zhang et al., "Generating informative and diverse conversational responses via adversarial information maximization", NeurIPS 2018]).
> >
> > However, we acknowledge that these metrics are far from perfect. Per your suggestion we have added the 10 generated outputs for all the methods on 4 examples which were selected randomly in Tables C.4-C.9. Additionally, we also set up a GPT-4-based evaluation to judge the coherence of the stories with the generated infills. Table C.3 presents the average rating based on coherence assigned by GPT-4 to the infills generated by each method. We observe that stories with infills sampled with the GFlowNet fine-tuned model achieve higher ratings than the baselines. We summarize the results here:
> >
> > | Method | GPT-4 Rating |
> > | ------ | ------------ |
> > | Prompting | 2.4 |
> > | Supervised FT | 2.7 |
> > | **GFlowNet FT** | 3.4 |
> > | ------ | ------------ |
> > | Reference | 4.3 |
> >
> > (Note that the reference infill's score should be taken as an upper bound, and the stories may have been present in GPT-4's training data.)
> >
> > > Limited discussion of the limitations of the proposed technique. Ultimately, the training method given here is a mostly-on-policy reinforcement learning method. A key challenge for such methods is exploration -- finding high-reward samples to reinforce. I would have appreciated more discussion of the sorts of posterior inference tasks that are and aren't likely solvable with the proposed techniques (at least without further innovations), possibly along with potential mitigations for these weaknesses.
> >
> > We will point out that the difficulty of exploration remains for hard problems. Our use of the LM to seed the replay buffer with potentially high-reward samples is one way to mitigate it. We will mention that tasks with a much larger latent space, e.g., theorem proving, will require strong exploration techniques to work.
> >
> >
> > > Around how long (in wall-clock time) does it take to LoRA fine-tune a GFlowNet on your tasks, e.g. for a 6B-parameter model?
> >
> > The arithmetic experiments take 24 hours, infilling experiments 12 hours, subjectivity experiments 6 hours, and sentence completion experiments 20 hours on a single 80GB NVIDIA A100 GPU. As with other RL-based fine-tuning approaches, GFlowNet fine-tuning is slower than supervised fine-tuning since it involves exploration. However, we do note that there are several places where our implementation can be sped up, and with multiple GPUs, the runtime can be improved significantly.
> >
> > > In Table 3, how were Test Accuracy numbers in the final row ("+ Supervised Fine-Tuning") generated? Were 10 samples of Z taken from the fine-tuned model, and aggregated via voting? Or are the Z samples still generated from q_{GFN} but now completed with Y drawn from the fine-tuned LM? Or is there no longer a voting procedure?
> >
> > The $Z$ samples are generated using $q_{GFN}$ and completed using the fine-tuned LM.

---

> > > ### Author Response · Authors · 2023-11-17
> > >
> > > > In principle, for tasks like story infilling, supervised fine-tuning (SFT) should be optimizing the same architecture as the GFN for an objective that has the same optimum (i.e., SFT is also a distribution-matching objective, where the optimum is the intractable posterior). Qualitatively, how do the samples from the SFT baseline look? (I think it would be nice to add them to Table B3 if possible!) If they are noticeably worse than the GFN samples, what would you attribute that to?
> > >
> > > Even in tasks like infilling, SFT learns the maximum likelihood solution rather than matching the reward distribution like GFlowNets do. This gives GFlowNet fine-tuning the advantage of better exploration of the solution space. We added some random samples from the baselines to Appendix C along with the GFlowNet samples. We observe that the baseline samples tend to be longer and continue beyond the infill region while GFlowNet samples are more concise and fit within the general style of the stories.
> > >
> > >
> > > > Also: for SFT and for the "just prompt the model to infill" baseline, do you start with the base language model or the reward language model that you fine-tuned with stories?
> > >
> > > The base model for this experiment is the GPT-2 model fine-tuned on stories, so the SFT and Prompting baselines both use this fine-tuned model.
> > >
> > >
> > > > You write on p23 that the reward model could often not distinguish between good and bad rationales. Does this mean that given a prompt (e.g.) Z="..., 1 + 4 = 5. The answer is:", the reward model assigns roughly equal probability to (a) the known correct answer Y from the training data (e.g., Y=14) and (b) the most-recently computed number (in this case, 5)? That's somewhat surprising to me!
> > >
> > > We will make that statement more clear. What we meant is that the language model is not good at scoring the Zs, i.e. an incorrect rationale has the same likelihood as a correct one. For example, for X = "Question: 1 + 0 - 1 =? Answer:", Y = "Therefore the answer is 0.", Z_1 = "1 + 0 = 1, 1 - 1 = 0" and Z_2 = "1 + 0 = 1, 1 + 1 = 2" we find that $p_{LM}(XZ_1Y) < p_{LM}(XZ_2Y)$. This is why we include some in-context examples for the reward.
> > >
> > > > At the bottom of p1, one of your citations is to a method that uses SMC, not MCMC, for which the notion of "mixing between modes" is not quite appropriate.
> > >
> > > We use the phrase "mixing between modes"  to refer to the general notion of modeling multi-modal distributions well. While SMC methods tend to perform better in sampling from multi-modal distributions, they can still suffer from "missing modes" in the generated samples.
> > >
> > > We hope we have addressed all of your concerns. Please don't hesitate to let us know if we can clarify anything else!

---

> > ### Author Response · Authors · 2023-11-19
> > **Replay buffer ablation**
> >
> > We ran additional experiments on SUBJ to ablate the effect of the replay buffer.
> >
> > | # of Samples / Method | GFlowNet fine-tuning | w/o buffer |
> > | --- | ----------| -------- |
> > | 10 | 71.4% | 61.6% |
> > | 20 | 81.1% | 70.3% |
> > | 50 | 87.7% | 59.0% |
> >
> > This confirms our intuition that on-policy or near-on-policy learning is not enough.

---

> > > ### Comment · Reviewer_H3x9 · 2023-11-23
> > >
> > > Thank you for the detailed responses! I continue to strongly support the acceptance of this paper.
> > >
> > > One note:
> > >
> > > > Even in tasks like infilling, SFT learns the maximum likelihood solution rather than matching the reward distribution like GFlowNets do.
> > >
> > > SFT with cross-entropy loss on ground-truth posterior samples maximizes $\mathbb{E}_{(x,z,y) \sim p}[\log q^\theta(z \mid x, y)] = \mathbb{E}[KL(p(z | x, y) || q^\theta(z | x, y))] + \text{const}$, i.e., it trains $q$ to match the posterior / reward distribution. (The expectation on the RHS is over $(x, y)$ pairs, but the loss is minimized when $q^\theta$ exactly matches the posterior for all $(x, y)$ pairs.)
> > >
> > > Typical RL objectives do train $q$ to *maximize expected reward*, i.e., to concentrate probability mass on the single MAP $z$ value. But SFT fits $q^\theta$ to the desired posterior, just as ordinary pre-training fits the LM to the data distribution (and not to spit out the "most likely sentence.")

---

### Official Review · Reviewer_ZdVU · 2023-11-05

**Soundness:** 2 fair
**Presentation:** 2 fair
**Contribution:** 3 good
**Rating:** 5
**Confidence:** 4

**Summary:**

Many applications of LLMs like text infilling and constrained generation requires probabilistic inference that is intractable for LLMs. E.g., for the task of text infilling we need to be able to compute the conditional probability p(text | prefix, suffix). The paper proposes to use tackle this problem by fine-tuning LMs with GFlowNet. Specifically, the author proposes to fine-tune LMs to approximate the desired conditional distribution, e.g., p(text | prefix, suffix) by mat. This paper conducted empirical evaluations on various benchmarks including text infilling and numerical reasoning.

**Strengths:**

Empirical results on synthetic arithmetic reasoning benchmarks seem to be very strong.

**Weaknesses:**

Overall the paper is hard to follow: the authors provide little background on reinforcement learning and GFlowNet training. In particular, the authors use many terminologies without/before defining them clearly, examples include “policy”, “reward”, “matching” a target distribution, “rewarding all valid integers equally leads to an expected gradient of zero for policy gradient methods.”

In section 2, by looking at the problem of using LLMs to generate random numbers between 0 - 100, the authors try to motivate the use of GFlowNet instead of PPO training. PPO training does not resolve the distribution skew because the reward function only considers whether the number lies between 0 - 100. One correct way to do it could be asking the LLM to generate a **sequence** of numbers sampled from 0 - 100 uniformly and assign a positive reward only if the frequency of the numbers are close to uniform. A major part of the introduction focuses on intractable posterior inference/conditional probabilities and the fact that Section 2 mentions nothing about them makes it hard to follow.

In section 3 the authors introduced some related problems in NLP that could potentially be solved by GFlowNet and in section 3.3 on page 5 that the authors finally describes GFlowNet and their training objective. What is the original subtrajectory balance objective? How do you modify it? What is the semantics of your objective function? Answer to these questions can help distinguish GFlowNet from other approaches from the methodology perspective.

Besides, some important related works of the field are missing from Section 3:
-for temperature scaling:
[1] leverages importance sampling to fine-tune LM p(x) such that it approximates the desired distribution p(x)^{1/T}. Their approach suffer from various problems such that the variance of loss is high due to the exponent 1/T. Given that the authors study this empirically, does the GFlowNet objective also suffer from this issue? If so, how is it resolved?

-for text infilling:
[2] and [3] both studies the problem of text infilling where [2] adopted a fine-tuning based approach. [6] and [7] tackles this problem by training insertion-based language models.

-for constrained generation:

`Current approaches to the problem use tokenwise approximations (Liu et al., 2021) or various problem-specific beam search and local search techniques`

Other than search-based approaches, frameworks like FUDGE [4] and NADO [5] trains auxiliary models (classifiers) and combine it with LMs to approximate the desired conditional distribution.

To summarize, GFlowNet seems to be a very very general framework that allows you to fine-tune an LM to approximate any distribution that is proportional to an arbitrary reward function r(x). Despite the experiment results showing advantages against vanilla baselines, the authors did not make a strong argument showing why GFlowNet would work better on these downstream tasks against existing approaches, including the ones mentioned above.

The main argument may be stronger/clearer if the authors focus more on the chain-of-thought reasoning part other than trying to provide a generic solution to all intractable inference for LMs.



[1] Shih, Andy, Dorsa Sadigh, and Stefano Ermon. "Long Horizon Temperature Scaling." arXiv preprint arXiv:2302.03686 (2023).

[2] Donahue, Chris, Mina Lee, and Percy Liang. "Enabling Language Models to Fill in the Blanks." Proceedings of the 58th Annual Meeting of the Association for Computational Linguistics. 2020.

[3] Zhu, Wanrong, Zhiting Hu, and Eric Xing. "Text infilling." arXiv preprint arXiv:1901.00158 (2019).

[4] Yang, Kevin, and Dan Klein. "FUDGE: Controlled Text Generation With Future Discriminators." Proceedings of the 2021 Conference of the North American Chapter of the Association for Computational Linguistics: Human Language Technologies. 2021.

[5] Meng, Tao, et al. "Controllable text generation with neurally-decomposed oracle." Advances in Neural Information Processing Systems 35 (2022): 28125-28139.

[6] Lu, Sidi, Tao Meng, and Nanyun Peng. "Insnet: An efficient, flexible, and performant insertion-based text generation model." Advances in Neural Information Processing Systems 35 (2022): 7011-7023.

[7] Susanto, R. H., Chollampatt, S., and Tan, L. Lexically constrained neural machine translation with levenshtein transformer. In Proceedings of the 58th Annual Meeting of the Association for Computational Linguistics (ACL), 2020.

**Questions:**

See above.

---

> ### Author Response · Authors · 2023-11-17
> **Thank you for your review!**
>
> We thank the reviewer for their detailed comments. We address each of them below. We have also updated the paper, with the changes colored red.
>
> As a general comment, we are aware that problem-specific models have been proposed for some of the tasks we consider, notably text infilling. (Thank you for suggesting the references.) However, our goal is not to train new models to achieve superior performance on any of these tasks but to extract knowledge from a **pretrained** LLM by efficiently fine-tuning it. To this end, we propose GFlowNet fine-tuning as a tool to solve general intractable inference problems in LLMs.
>
> > Overall the paper is hard to follow: the authors provide little background on reinforcement learning and GFlowNet training. In particular, the authors use many terminologies without/before defining them clearly, examples include “policy”, “reward”, “matching” a target distribution, “rewarding all valid integers equally leads to an expected gradient of zero for policy gradient methods.”
>
> We appreciate the feedback. While we cannot provide a comprehensive background on RL and GFlowNets due to space constraints, we have added pointers to the relevant literature and a glossary in Appendix A.
>
> > In section 2, by looking at the problem of using LLMs to generate random numbers between 0 - 100, the authors try to motivate the use of GFlowNet instead of PPO training. PPO training does not resolve the distribution skew because the reward function only considers whether the number lies between 0 - 100.
>
> We would like to clarify that the problem considered in section 2 serves as a simple demonstration of the underlying problem of amortized inference, where one has access to a likelihood and the problem is to sample from the desired distribution. RL algorithms such as PPO are formulated to maximize the reward, and thus fail in this scenario. Note that the reward used is the same for GFlowNets and PPO, and the only thing that differs is the fine-tuning algorithm. This example serves **precisely** to illustrate that reward maximization (PPO) is less appropriate than distribution matching (GFlowNet) in some settings, motivating our approach.
>
> > One correct way to do it could be asking the LLM to generate a sequence of numbers sampled from 0 - 100 uniformly and assign a positive reward only if the frequency of the numbers is close to uniform.
>
> Our goal here is not to sample random numbers from LLMs but to demonstrate a critical shortcoming with existing fine-tuning paradigms. The approach of rewarding a sequence of numbers is undesirable for many reasons: 1) PPO can collapse to deterministically generating a sequence of uniformly distributed **but not independent** numbers (e.g., just listing the numbers from 0 to 100 in order), which would not yield a reliable sampler; 2) it does not scale to scenarios where we want to sample long reasoning chains, because we will need to generate many of these long chains before receiving a reward; 3) the reward is hard to design and may have a high variance.
>
> > A major part of the introduction focuses on intractable posterior inference/conditional probabilities and the fact that Section 2 mentions nothing about them makes it hard to follow.
>
> We have made the transition more clear. Section 2, through a minimal problem of using language models to sample from a distribution given an unnormalized density, demonstrates the shortcomings of reward-maximizing RL for posterior inference and introduces GFlowNet fine-tuning as the appropriate tool.

---

> ### Author Response · Authors · 2023-11-17
>
> > In section 3 the authors introduced some related problems in NLP that could potentially be solved by GFlowNet and in section 3.3 on page 5 that the authors finally describes GFlowNet and their training objective. What is the original subtrajectory balance objective? How do you modify it? What is the semantics of your objective function? Answers to these questions can help distinguish GFlowNet from other approaches from the methodology perspective.
>
> We limited our exposition of GFlowNets in general to maintain the focus on the setting studied in the paper and direct the reader to relevant prior work. Specifically, the original subtrajectory balance term for a subtrajectory $\tau_{m:n} = s_m\rightarrow\dots\rightarrow s_n$ is:
>
> $$ℒ_{\text{subTB} }(\tau_{m:n}) = \bigg[\log\frac{F(s_m)\prod_{i=m}^{n-1}P_F(s_{i+1}\mid s_i)}{F(s_n)\prod_{i=m}^{n-1}P_B(s_i \mid s_{i+1})}\bigg]^2$$
>
> where $P_F$ is the sampling policy -- called $q_{\text{policy}}$ in our paper -- and $P_B$ is a "backward policy" (we refer to past GFlowNet work for discussion of what this means, but note that in our setting the $P_B$ terms are always 1 and can be ignored).
>
> Our modification leverages an important aspect of the problem setting: During generation, the trajectory can terminate at each state. To account for this, we incorporate the modification proposed by [Deleu et al., 2022], which involves incorporating the termination likelihood and the reward at each state. Specifically, using the fact that at convergence we have $R(s_n^\top)=F(s_n)P_F(\top\mid s_n)$, we simply substitute $R(s_n^\top)/P_F(\top\mid s_n)$ for $F(s_n)$ in the loss above. Rearrangement of the terms yields exactly the term being summed in our loss (3).
>
> These modifications together allow us to parameterize the GFlowNet through only the forward policy, avoiding the need to train additional estimators.
>
> GFlowNet objectives have been studied extensively in prior work. Semantically, these learning objectives aim to satisfy constraints on the distribution over the trajectories given by the policy to sample proportionally to the reward.
>
> > Besides, some important related works of the field are missing from Section 3
>
> Thanks for pointing to the additional related work! We have added the missing references.
>
> > for temperature scaling: [1] leverages importance sampling to fine-tune LM p(x) such that it approximates the desired distribution p(x)^{1/T}. Their approach suffers from various problems such that the variance of loss is high due to the exponent 1/T. Given that the authors study this empirically, does the GFlowNet objective also suffer from this issue? If so, how is it resolved?
>
> A key advantage of our method is that we do not rely on importance sampling for off-policy learning, an advantageous property of GFlowNets that was studied by [Malkin et al., 2023]. The variance of our loss is therefore not prohibitive for stable training. Independent from importance sampling, the variance of our loss does increase with lower temperatures (especially early in training) simply because this increases the difference between the target distribution and the initial LM distribution that we are fine-tuning. We mitigate this by slowly annealing the temperature from $1$ down to its final value throughout training. As a result, at any given point during training, the GFlowNet distribution is close to the current target distribution and the variance (and magnitude) of the loss is small. Empirically, we found that this greatly reduces the variance of our loss to the point where it is not a significant concern.
>
> Overall, we would like to reiterate that the key goal of the paper is not to achieve state-of-the-art performance in each of the settings we consider. Rather, the GFlowNet fine-tuning paradigm, as you point out in your review, provides a unified view for all of these intractable inference problems and proposes a single general approach to tackle all of these problems. The breadth of our experiments intends to demonstrate the problem-agnostic nature of GFlowNet fine-tuning. Different problems simply correspond to different reward functions.

---

> > ### Author Response · Authors · 2023-11-20
> > **Follow-up**
> >
> > Dear Reviewer ZdVU,
> >
> > Thank you again for your review. We have posted responses to your questions and comments above. Could you please let us know if they have affected your assessment of the paper and if you have any more questions before the end of the rebuttal period? We would be happy to provide any further clarification.
> >
> > Thank you,
> >
> > The authors.

---

> > > ### Comment · Reviewer_ZdVU · 2023-11-22
> > >
> > > Thank you for your clarification about the motivating example (Section 2). I am now convinced that RL's standard reward maximization objective cannot be formulated to simulate the objective of matching a specific target distribution.
> > >
> > > Thank you for answering my questions about the loss function in Section 3 and providing extra details, which really helped me understand this paper better. Even though prior works might have extensively discussed the loss of GFlowNet, I believe that some minimal technical background is needed for readers who are not familiar with GFlowNet, not just to educate them but also help them understand why your approach should work better in certain scenarios.
> > >
> > > I would like to say that I do value this work in the sense that it presents GFlowNet as a general approach for tackling the problem of intractable inference with LLMs. However, just providing a general framework is not enough: the important part is to discuss the advantages and limitations of the approach. Consider an extreme case, I can also claim that naive MCMC sampling is a general framework that solves everything, but it doesn't, because of high computation cost. I'm very willing to believe that "being a general framework" is not the only advantage of GFlowNet, but the other advantages are not clear to me.
> > > Given a particular application scenario like constrained generation, should I use GFlowNet or not? The answer is unclear to me. You referred to some prior works but it is not clear from your work. At least giving some discussion of GFlowNet's limitations could also help; in author response,  the discussion on the variance of the GFlowNet objective serves this purpose: so now I know that it can suffer from high variance when the temperature is low and you can mitigate this by gradually decreasing the temperature from 1.0.
> > >
> > > I would be willing to increase my score if I see a stronger argument for the advantages of GFlowNet in intractable inference with LLMs or a more comprehensive discussion of the limitations of GFlowNet.

---

> ### Author Response · Authors · 2023-11-22
> **Advantages and limitations**
>
> We appreciate your engagement and additional comments.
>
> We see the advantages of our approach to intractable inference with LLMs as the following:
> - **Principled objectives:** Ours is **the first attempt at general-purpose amortized inference in LLMs** that has a guarantee of matching the target distribution when trained to zero loss. As such, it has relatively few "moving parts" beyond those present in the design of any RL algorithm. This is in contrast to prior work that has used Monte Carlo approaches or specialized wake-sleep algorithms for specific intractable inference problems, which, respectively, do not perform amortization and do not feature a loss that can be optimized to zero to yield an exact sampler.
> - **Bayesian formulation:**  We offer a **clean Bayesian inference perspective on infilling and chain-of-thought reasoning** and present an algorithm to perform this inference. This is already a major advantage over the standard approach to chain-of-thought reasoning through judicious prompting and in-context learning, in which a Bayesian formulation emerges only in post-hoc analysis (cf. the recent literature on the Bayesian interpretations of ICL).
> - **Versatility and breadth of applications:** As you noted, we show that our approach can be used to fine-tune LLMs to solve **a wide variety of intractable inference problems**, namely infilling, chain-of-thought / reasoning chain inference, tool use, and even the fundamental problem of tempered autoregressive sampling. The versatility of GFlowNet fine-tuning is a key advantage over approaches that focus on specific inference problems.
>
> Thus, our general answer is that "if you can afford to GFlowNet-fine-tune your LLM for your inference problem, then you should try to do so".
>
> This leads naturally to the question of limitations, some of which we have already discussed some in our paper and responses. The three main ones we see are:
> - **Compute cost:** GFlowNet fine-tuning is more expensive than supervised fine-tuning, as it involves exploration. (This can in part be mitigated by seeding of the replay buffer with high-quality samples obtained using a different algorithm.)
> - **Sensitivity to training parameters:** GFlowNets are reinforcement learning algorithms and as such require choices of exploration parameters in addition to those present in any fine-tuning setting. (The use of the replay buffer and the temperature annealing schedule are especially important, as we have found.) The need to search for good parameters can be costly, and we have not explored the full range of possible settings and tricks.
> - **Reliance on reward model and possible misalignment:** GFlowNet fine-tuning requires a fixed reward model (or a lightly varying one in the case of an EM loop, but in any case defined by a base LM). While formulating the reward as an unnormalized Bayesian posterior is straightforward in cases such as infilling, it may be difficult to balance quality and diversity in general constrained generation settings. Additionally, as we allude to in the conclusion and in Appendix E, high-reward sequences are not necessarily of high quality. To summarize, while GFlowNet fine-tuning seems to be quite good at learning to sample a posterior, it does not address the question of what that posterior should be, nor does it address the failures of the LLM reward model to capture the desired properties.
>
> We hope that this discussion helps put our contributions into perspective.

---

> > ### Comment · Reviewer_ZdVU · 2023-11-22
> >
> > Thank you for your clarification. I have updated my score. I hope this discussion can be extended with further details/evidence and included in your main text.

---

> > ### Comment · Reviewer_H3x9 · 2023-11-23
> > **Comment from a different reviewer**
> >
> > I am not the writer of this review, but I wanted to follow up on part of your response.
> >
> > > specialized wake-sleep algorithms for specific intractable inference problems, which ... do not feature a loss that can be optimized to zero to yield an exact sampler.
> >
> > I am not sure which prior work you are referring to here, but in general, don't wake-sleep algorithms often optimize a KL divergence that is also minimized by an exact sampler?
> >
> > > Compute cost: GFlowNet fine-tuning is more expensive than supervised fine-tuning, as it involves exploration. (This can in part be mitigated by seeding of the replay buffer with high-quality samples obtained using a different algorithm.)
> >
> > This is one way to frame it, but I think it's important to point out that purely on-policy learning in your framework has the same limitation that many naive Monte Carlo schemes (e.g. likelihood weighting or rejection sampling) have: if the posterior is very different from the prior / initial policy, the exploration will take exponentially long (roughly, exponential in the KL divergence between prior and posterior) to find good posterior samples to reinforce.
> >
> > (I really like this paper, but I do think it's important to acknowledge both the existence of alternative approaches to amortized inference & the possible limitations of this approach.)

---

### Meta-Review · Area_Chair_z1Ta · 2023-12-09

**Metareview:**

Many applications of LLMs like text infilling and constrained generation can be viewed as sampling latent variables from a posterior distribution in large language models (LLMs), here the latent variables might take the form of prompts, reasoning chains, etc. This paper presents a new approach to this intractable inference problem, by finetuning LLMs with GFlowNet for amortized inference. Specifically, the approach samples a sequence of tokens via a sequence of constructive steps, with a probability proportional to a reward function (the product of the likelihood and the prior, leading to the joint distribution). Such a Bayesian inference method is different from MLE-based fine-tuning and reward-maximization-based fine-tuning, which tend to make the learned distribution more concentrated on one or few modes, potentially leading to incorrect outputs. Experimental results show the effectiveness of GFlowNets-based fine-tuning in improving text generation and reasoning tasks. Reviewers have found the work is novel and interesting. More discussion on the limitations of the method (e.g., as an on-policy approach) and its relationship to possible alternatives is desirable.

**Justification For Why Not Higher Score:**

NA

**Justification For Why Not Lower Score:**

This work is novel and interesting.

---

### Decision · Program_Chairs · 2024-01-16

Accept (oral)